# ViDT: An Efficient and Effective Fully Transformer-based Object Detector

**Hwanjun Song**[1], **Deqing Sun**[2], **Sanghyuk Chun**[1], **Varun Jampani**[2], **Dongyoon Han**[1],
**Byeongho Heo**[1], **Wonjae Kim**[1], **Ming-Hsuan Yang**[2,3,4]
[1]NAVER AI Lab  [2]Google Research  [3]University of California at Merced  [4]Yonsei University
{hwanjun.song, sanghyuk.c, dongyoon.han, bh.heo, wonjae.kim}@navercorp.com
{deqingsun, varunjampani}@google.com, mhyang@ucmerced.edu

## Abstract

Transformers are transforming the landscape of computer vision, especially for recognition tasks. Detection transformers are the first fully end-to-end learning systems for object detection, while vision transformers are the first fully transformer-based architecture for image classification. In this paper, we integrate Vision and Detection Transformers (ViDT) to build an effective and efficient object detector. ViDT introduces a reconfigured attention module to extend the recent Swin Transformer to be a standalone object detector, followed by a computationally efficient transformer decoder that exploits multi-scale features and auxiliary techniques essential to boost the detection performance without much increase in computational load. Extensive evaluation results on the Microsoft COCO benchmark dataset demonstrate that ViDT obtains the best AP and latency trade-off among existing fully transformer-based object detectors, and achieves **49.2**AP owing to its high scalability for large models. We release the code and trained models at https://github.com/naver-ai/vidt.

## 1 Introduction

Object detection is the task of predicting both bounding boxes and object classes for each object of interest in an image. Modern deep object detectors heavily rely on meticulously designed components, such as anchor generation and non-maximum suppression (Papageorgiou & Poggio, 2000; Liu et al., 2020). As a result, the performance of these object detectors depend on specific postprocessing steps, which involve complex pipelines and make fully end-to-end training difficult.

Motivated by the recent success of Transformers (Vaswani et al., 2017) in NLP, numerous studies introduce Transformers into computer vision tasks. Carion et al. (2020) proposed **Detection Transformers (DETR)** to eliminate the meticulously designed components by employing a simple transformer encoder and decoder architecture, which serves as a neck component to bridge a CNN body for feature extraction and a detector head for prediction. Thus, DETR enables end-to-end training of deep object detectors. By contrast, Dosovitskiy et al. (2021) showed that a fully-transformer backbone without any convolutional layers, **Vision Transformer (ViT)**, achieves the state-of-the-art results in image classification benchmarks. Approaches like ViT have been shown to learn effective representation models without strong human inductive biases, e.g., meticulously designed components in object detection (DETR), locality-aware designs such as convolutional layers and pooling mechanisms. However, there is a lack of effort to synergize DETR and ViT for a better object detection architecture. In this paper, we integrate both approaches to build a fully transformer-based, end-to-end object detector that achieves state-of-the-art performance without increasing computational load.

A straightforward integration of DETR and ViT can be achieved by replacing the ResNet backbone

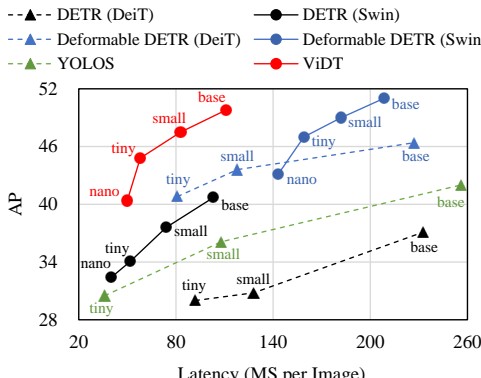

Figure 1. AP and latency (milliseconds) summarized in Table 2. The text in the plot indicates the backbone model size.

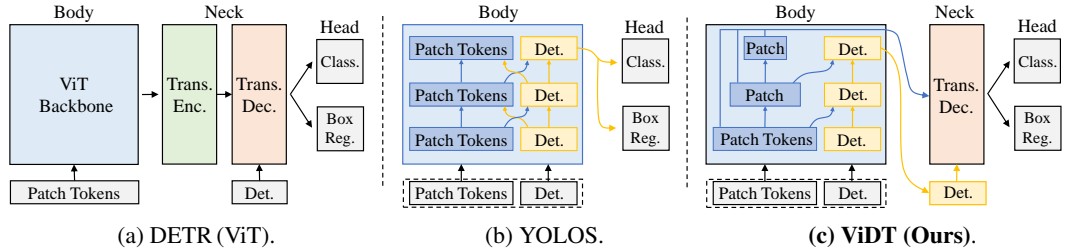

(a) DETR (ViT).     (b) YOLOS.     (c) ViDT (Ours).

Figure 2. Pipelines of fully transformer-based object detectors. DETR (ViT) means Detection Transformer that uses ViT as its body. The proposed ViDT synergizes DETR (ViT) and YOLOS and achieves best AP and latency trade-off among fully transformer-based object detectors.

(body) of DETR with ViT – Figure 2(a). This naive integration, **DETR (ViT)**[1], has two limitations. First, the canonical ViT suffers from the quadratic increase in complexity w.r.t. image size, resulting in the lack of scalability. Furthermore, the attention operation at the transformer encoder and decoder (i.e., the "neck" component) adds significant computational overhead to the detector. Therefore, the naive integration of DETR and ViT show very high latency – the blue lines of Figure 1.

Recently, Fang et al. (2021) propose an extension of ViT to object detection, named **YOLOS**, by appending the detection tokens [DET] to the patch tokens [PATCH] (Figure 2(b)), where [DET] tokens are learnable embeddings to specify different objects to detect. YOLOS is a neck-free architecture and removes the additional computational costs from the neck encoder. However, YOLOS shows limited performance because it cannot use additional optimization techniques on the neck architecture, e.g., multi-scale features and auxiliary loss. In addition, YOLOS can only accommodate the canonical transformer due to its architectural limitation, resulting in a quadratic complexity w.r.t. the input size.

In this paper, we propose a novel integration of **Vi**sion and **D**etection **T**ransformers (**ViDT**) (Figure 2(c)). Our contributions are three-folds. First, ViDT introduces a modified attention mechanism, named Reconfigured Attention Module (RAM), that facilitates any ViT variant to handle the appended [DET] and [PATCH] tokens for object detection. Thus, we can modify the latest Swin Transformer (Liu et al., 2021) backbone with RAM to be an object detector and obtain high scalability using its local attention mechanism with linear complexity. Second, ViDT adopts a lightweight encoder-free neck architecture to reduce the computational overhead while still enabling the additional optimization techniques on the neck module. Note that the neck encoder is unnecessary because RAM directly extracts fine-grained representation for object detection, i.e., [DET] tokens. As a result, ViDT obtains better performance than neck-free counterparts. Finally, we introduce a new concept of token matching for knowledge distillation, which brings additional performance gains from a large model to a small model without compromising detection efficiency.

ViDT has two architectural advantages over existing approaches. First, similar to YOLOS, ViDT takes [DET] tokens as the additional input, maintaining a fixed scale for object detection, but constructs hierarchical representations starting with small-sized image patches for [PATCH] tokens. Second, ViDT can use the hierarchical (multi-scale) features and additional techniques without a significant computation overhead. Therefore, as a fully transformer-based object detector, ViDT facilitates better integration of vision and detection transformers. Extensive experiments on Microsoft COCO benchmark (Lin et al., 2014) show that ViDT is highly scalable even for large ViT models, such as Swin-base with 0.1 billion parameters, and achieves the best AP and latency trade-off.

## 2 PRELIMINARIES

**Vision transformers** process an image as a sequence of small-sized image patches, thereby allowing all the positions in the image to interact in attention operations (i.e., global attention). However, the canonical ViT (Dosovitskiy et al., 2021) is not compatible with a broad range of vision tasks due to its high computational complexity, which increases quadratically with respect to image size. The Swin Transformer (Liu et al., 2021) resolves the complexity issue by introducing the notion of shifted windows that support local attention and patch reduction operations, thereby improving compatibility for dense prediction task such as object detection. A few approaches use vision transformers as detector backbones but achieve limited success (Heo et al., 2021; Fang et al., 2021).

---

[1]We refer to each model based on the combinations of its body and neck. For example, DETR (DeiT) indicates that DeiT (vision transformers) is integrated with DETR (detection transformers).

**Detection transformers**  eliminate the meticulously designed components (e.g., anchor generation and non-maximum suppression) by combining convolutional network backbones and Transformer encoder-decoders. While the canonical DETR (Carion et al., 2020) achieves high detection performance, it suffers from very slow convergence compared to previous detectors. For example, DETR requires 500 epochs while the conventional Faster R-CNN (Ren et al., 2015) training needs only 37 epochs (Wu et al., 2019). To mitigate the issue, Zhu et al. (2021) propose Deformable DETR which introduces deformable attention for utilizing multi-scale features as well as expediting the slow training convergence of DETR. In this paper, we use the Deformable DETR as our base detection transformer framework and integrate it with the recent vision transformers.

**DETR (ViT)**  is a straightforward integration of DETR and ViT, which uses ViT as a feature extractor, followed by the transformer encoder-decoder in DETR. As illustrated in Figure 2(a), it is a *body–neck–head* structure; the representation of input [PATCH] tokens are extracted by the ViT backbone and then directly fed to the transformer-based encoding and decoding pipeline. To predict multiple objects, a fixed number of learnable [DET] tokens are provided as additional input to the decoder. Subsequently, output embeddings by the decoder produce final predictions through the detection heads for classification and box regression. Since DETR (ViT) does not modify the backbone at all, it can be flexibly changed to any latest ViT model, e.g., Swin Transformer. Additionally, its neck decoder facilitates the aggregation of multi-scale features and the use of additional techniques, which help detect objects of different sizes and speed up training (Zhu et al., 2021). However, the attention operation at the neck encoder adds significant computational overhead to the detector. In contrast, ViDT resolves this issue by directly extracting fine-grained [DET] features from Swin Transformer with RAM without maintaining the transformer encoder in the neck architecture.

**YOLOS** (Fang et al., 2021)  is a canonical ViT architecture for object detection with minimal modifications. As illustrated in Figure 2(b), YOLOS achieves a *neck-free* structure by appending randomly initialized learnable [DET] tokens to the sequence of input [PATCH] tokens. Since all the embeddings for [PATCH] and [DET] tokens interact via global attention, the final [DET] tokens are generated by the fine-tuned ViT backbone and then directly generate predictions through the detection heads without requiring any neck layer. While the naive DETR (ViT) suffers from the computational overhead from the neck layer, YOLOS enjoys efficient computations by treating the [DET] tokens as additional input for ViT. YOLOS shows that 2D object detection can be accomplished in a pure sequence-to-sequence manner, but this solution entails *two* inherent limitations:

1) YOLOS inherits the drawback of the canonical ViT; the high computational complexity attributed to the global attention operation. As illustrated in Figure 1, YOLOS shows very poor latency compared with other fully transformer-based detectors, especially when its model size becomes larger, i.e., small $\rightarrow$ base. Thus, YOLOS is *not scalable* for the large model.

2) YOLOS cannot benefit from using additional techniques essential for better performance, e.g., multi-scale features, due to the absence of the neck layer. Although YOLOS used the same DeiT backbone with Deformable DETR (DeiT), its AP was lower than the straightforward integration.

In contrast, the encoder-free neck architecture of ViDT enjoys the additional optimization techniques from Zhu et al. (2021), resulting in the faster convergence and the better performance. Further, our RAM enables to combine Swin Transformer and the sequence-to-sequence paradigm for detection.

## 3   ViDT: Vision and Detection Transformers

ViDT first reconfigures the attention model of Swin Transformer to support standalone object detection while fully reusing the parameters of Swin Transformer. Next, it incorporates an encoder-free neck layer to exploit multi-scale features and two essential techniques: auxiliary decoding loss and iterative box refinement. We further introduce knowledge distillation with token matching to benefit from large ViDT models.

### 3.1   Reconfigured Attention Module

Applying patch reduction and local attention scheme of Swin Transformer to the sequence-to-sequence paradigm is challenging because (1) the number of [DET] tokens must be maintained at a fixed-scale and (2) the lack of locality between [DET] tokens. To address this challenge, we introduce a reconfigured attention module (RAM)[2] that decomposes a single global attention associated with [PATCH] and [DET] tokens into the *three* different attention, namely [PATCH] $\times$ [PATCH], [DET] $\times$ [DET],

---

[2]This reconfiguration scheme can be easily applied to other ViT variants with simple modification.

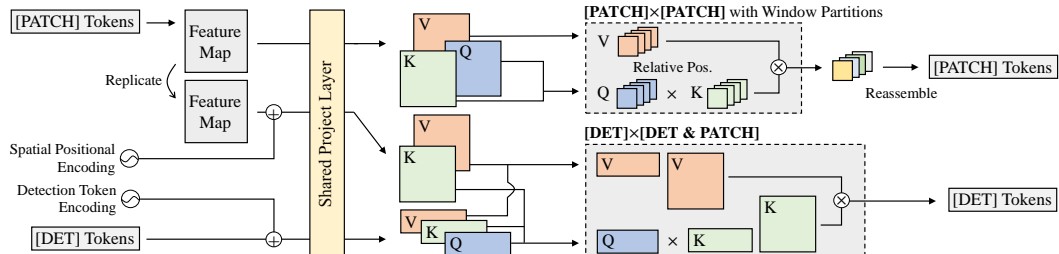

Figure 3. Reconfigured Attention Module (Q: query, K: key, V: value). The skip connection and feedforward networks following the attention operation is omitted just for ease of exposition.

and [DET] × [PATCH] attention. Based on the decomposition, the efficient schemes of Swin Transformer are applied only to [PATCH] × [PATCH] attention, which is the heaviest part in computational complexity, without breaking the two constraints on [DET] tokens. As illustrated in Figure 3, these modifications fully reuse all the parameters of Swin Transformer by sharing projection layers for [DET] and [PATCH] tokens, and perform the three different attention operations:

- [PATCH] × [PATCH] Attention: The initial [PATCH] tokens are progressively calibrated across the attention layers such that they aggregate the key contents in the global feature map (i.e., spatial form of [PATCH] tokens) according to the attention weights, which are computed by ⟨query, key⟩ pairs. For [PATCH]×[PATCH] attention, Swin Transformer performs local attention on each window partition, but its shifted window partitioning in successive blocks bridges the windows of the preceding layer, providing connections among partitions to capture global information. Without modifying this concept, we use the same policy to generate hierarchical [PATCH] tokens. Thus, the number of [PATCH] tokens is reduced by a factor of 4 at each stage; the resolution of feature maps decreases from $H/4 \times W/4$ to $H/32 \times W/32$ over a total of four stages, where $H$ and $W$ denote the width and height of the input image, respectively.

- [DET] × [DET] Attention: Like YOLOS, we append one hundred learnable [DET] tokens as the additional input to the [PATCH] tokens. As the number of [DET] tokens specifies the number of objects to detect, their number must be maintained with a fixed-scale over the transformer layers. In addition, [DET] tokens do not have any locality unlike the [PATCH] tokens. Hence, for [DET] × [DET] attention, we perform global self-attention while maintaining the number of them; this attention helps each [DET] token to localize a different object by capturing the relationship between them.

- [DET] × [PATCH] Attention: This is cross-attention between [DET] and [PATCH] tokens, which produces an object embedding per [DET] token. For each [DET] token, the key contents in [PATCH] tokens are aggregated to represent the target object. Since the [DET] tokens specify different objects, it produces different object embeddings for diverse objects in the image. Without the cross-attention, it is infeasible to realize the standalone object detector. As shown in Figure 3, ViDT binds [DET] × [DET] and [DET] × [PATCH] attention to process them at once to increase efficiency.

We replace all the attention modules in Swin Transformer with the proposed RAM, which receives [PATCH] and [DET] tokens (as shown in "Body" of Figure 2(c)) and then outputs their calibrated new tokens by performing the three different attention operations in parallel.

**Positional Encoding.** ViDT adopts different positional encodings for different types of attention. For [PATCH]×[PATCH] attention, we use the relative position bias (Hu et al., 2019) originally used in Swin Transformer. In contrast, the learnable positional encoding is added for [DET] tokens for [DET]×[DET] attention because there is no particular order between [DET] tokens. However, for [DET] × [PATCH] attention, it is crucial to inject *spatial bias* to the [PATCH] tokens due to the permutation-equivariant in transformers, ignoring spatial information of the feature map. Thus, ViDT adds the sinusoidal-based spatial positional encoding to the feature map, which is reconstructed from the [PATCH] tokens for [DET] × [PATCH] attention, as can be seen from the left side of Figure 3. We present a thorough analysis of various spatial positional encodings in Section 4.2.1.

**Use of [DET] × [PATCH] Attention.** Applying cross-attention between [DET] and [PATCH] tokens adds additional computational overhead to Swin Transformer, especially when it is activated at the bottom layer due to the large number of [PATCH] tokens. To minimize such computational overhead, ViDT only activates the cross-attention at the last stage (the top level of the pyramid) of Swin Transformer, which consists of two transformer layers that receives [PATCH] tokens of size $H/32 \times W/32$.

Thus, only self-attention for [DET] and [PATCH] tokens are performed for the remaining stages except the last one. In Section 4.2.2 we show that this design choice helps achieve the highest FPS, while achieving similar detection performance as when cross-attention is enabled at every stage. We provide more details on RAM including its complexity analysis and algorithmic design in Appendix A.

## 3.2 ENCODER-FREE NECK STRUCTURE

To exploit multi-scale feature maps, ViDT incorporates a decoder of multi-layer deformable transformers (Zhu et al., 2021). In the DETR family (Figure 2(a)), a transformer encoder is required at the neck to transform features extracted from the backbone for image classification into the ones suitable for object detection; the encoder is generally computationally expensive since it involves [PATCH] × [PATCH] attention. However, ViDT maintains only a transformer decoder as its neck, in that Swin Transformer with RAM directly extracts fine-grained features suitable for object detection as a standalone object detector. Thus, the neck structure of ViDT is computationally efficient.

The decoder receives two inputs from Swin Transformer with RAM: (1) [PATCH] tokens generated from each stage (i.e., four multi-scale feature maps, $\{x^l\}_{l=1}^L$ where $L = 4$) and (2) [DET] tokens generated from the last stage. The overview is illustrated in "Neck" of Figure 2(c). In each deformable transformer layer, [DET] × [DET] attention is performed first. For each [DET] token, multi-scale deformable attention is applied to produce a new [DET] token, aggregating a small set of key contents sampled from the multi-scale feature maps $\{x^l\}_{l=1}^L$,

$$\text{MSDeformAttn}([\text{DET}], \{x^l\}_{l=1}^L) = \sum_{m=1}^M W_m \Big[ \sum_{l=1}^L \sum_{k=1}^K A_{mlk} \cdot W_m' x^l \big(\phi_l(p) + \Delta p_{mlk}\big)\Big], \quad (1)$$

where $m$ indices the attention head and $K$ is the total number of sampled keys for content aggregation. In addition, $\phi_l(p)$ is the reference point of the [DET] token re-scaled for the $l$-th level feature map, while $\Delta p_{mlk}$ is the sampling offset for deformable attention; and $A_{mlk}$ is the attention weights of the $K$ sampled contents. $W_m$ and $W_m'$ are the projection matrices for multi-head attention.

**Auxiliary Techniques for Additional Improvements.** The decoder of ViDT follows the standard structure of multi-layer transformers, generating refined [DET] tokens at each layer. Hence, ViDT leverages the two auxiliary techniques used in (Deformable) DETR for additional improvements:

- Auxiliary Decoding Loss: Detection heads consisting of two feedforward networks (FNNs) for box regression and classification are attached to every decoding layer. All the training losses from detection heads at different scales are added to train the model. This helps the model output the correct number of objects without non-maximum suppression (Carion et al., 2020).

- Iterative Box Refinement: Each decoding layer refines the bounding boxes based on predictions from the detection head in the previous layer. Therefore, the box regression process progressively improves through the decoding layers (Zhu et al., 2021).

These two techniques are essential for transformer-based object detectors because they significantly enhance detection performance without compromising detection efficiency. We provide an ablation study of their effectiveness for object detection in Section 4.3.1.

## 3.3 KNOWLEDGE DISTILLATION WITH TOKEN MATCHING FOR OBJECT DETECTION

While a large model has a high capacity to achieve high performance, it can be computationally expensive for practical use. As such, we additionally present a simple knowledge distillation approach that can transfer knowledge from the large ViDT model by token matching. Based on the fact that all ViDT models has exactly the same number of [PATCH] and [DET] tokens regardless of their scale, a small ViDT model (a student model) can easily benefit from a pre-trained large ViDT (a teacher model) by matching its tokens with those of the large one, thereby bringing out higher detection performance at a lower computational cost.

Matching all the tokens at every layer is very inefficient in training. Thus, we only match the tokens contributing the most to prediction. The two sets of tokens are directly related: (1) $\mathcal{P}$: the set of [PATCH] tokens used as multi-scale feature maps, which are generated from each stage in the body, and (2) $\mathcal{D}$: the set of [DET] tokens, which are generated from each decoding layer in the neck. Accordingly, the distillation loss based on token matching is formulated by

$$\ell_{dis}(\mathcal{P}_s, \mathcal{D}_s, \mathcal{P}_t, \mathcal{D}_t) = \lambda_{dis} \Big( \frac{1}{|\mathcal{P}_s|} \sum_{i=1}^{|\mathcal{P}_s|} \Big\| \mathcal{P}_s[i] - \mathcal{P}_t[i] \Big\|_2 + \frac{1}{|\mathcal{D}_s|} \sum_{i=1}^{|\mathcal{D}_s|} \Big\| \mathcal{D}_s[i] - \mathcal{D}_t[i] \Big\|_2 \Big), \quad (2)$$

| Backbone | Type (Size) | Train Data | Epochs | Resolution | Params | ImageNet Acc. |
|---|---|---|---|---|---|---|
| | DeiT-tiny (🏃) | ImageNet-1K | 300 | 224 | 6M | 74.5 |
| DeiT | DeiT-small (🏃) | ImageNet-1K | 300 | 224 | 22M | 81.2 |
| | DeiT-base (🏃) | ImageNet-1K | 300 | 384 | 87M | 85.2 |
| | Swin-nano | ImageNet-1K | 300 | 224 | 6M | 74.9 |
| Swin | Swin-tiny | ImageNet-1K | 300 | 224 | 28M | 81.2 |
| Transformer | Swin-small | ImageNet-1K | 300 | 224 | 50M | 83.2 |
| | Swin-base | ImageNet-22K | 90 | 224 | 88M | 86.3 |

Table 1. Summary on the ViT backbone. "🏃" is the distillation strategy for classification (Touvron et al., 2021).

where the subscripts $s$ and $t$ refer to the student and teacher model. $\mathcal{P}[i]$ and $\mathcal{D}[i]$ return the $i$-th [PATCH] and [DET] tokens, $n$-dimensional vectors, belonging to $\mathcal{P}$ and $\mathcal{D}$, respectively. $\lambda_{dis}$ is the coefficient to determine the strength of $\ell_{dis}$, which is added to the detection loss if activated.

## 4 EVALUATION

In this section, we show that ViDT achieves the best trade-off between accuracy and speed (Section 4.1). Then, we conduct detailed ablation study of the reconfigured attention module (Section 4.2) and additional techniques to boost detection performance (Section 4.3). Finally, we provide a complete analysis of all components available for ViDT (Section 4.4).

**Dataset.** We carry out object detection experiments on the Microsoft COCO 2017 benchmark dataset (Lin et al., 2014). All the fully transformer-based object detectors are trained on 118K training images and tested on 5K validation images following the literature (Carion et al., 2020).

**Algorithms.** We compare ViDT with two existing fully transformer-based object detection pipelines, namely DETR (ViT) and YOLOS. Since DETR (ViT) follows the general pipeline of (Deformable) DETR by replacing its ResNet backbone with other ViT variants; hence, we use one canonical ViT and one latest ViT variant, DeiT and Swin Transformer, as its backbone without any modification. In contrast, YOLOS is the canonical ViT architecture, thus only DeiT is available. Table 1 summarizes all the ViT models pre-trained on ImageNet used for evaluation. Note that publicly available pre-trained models are used except for Swin-nano. We newly configure Swin-nano[3] comparable to DeiT-tiny, which is trained on ImageNet with the identical setting. Overall, with respect to the number of parameters, Deit-tiny, -small, and -base are comparable to Swin-nano, -tiny, and -base, respectively. Please see Appendix B.2 for the detailed pipeline of compared detectors.

**Implementation Details.** All the algorithms are implemented using PyTorch and executed using eight NVIDIA Tesla V100 GPUs. We train ViDT using AdamW (Loshchilov & Hutter, 2019) with the same initial learning rate of $10^{-4}$ for its body, neck and head. In contrast, following the (Deformable) DETR setting, DETR (ViT) is trained with the initial learning rate of $10^{-5}$ for its pre-trained body (ViT backbone) and $10^{-4}$ for its neck and head. YOLOS and ViDT (w.o. Neck) are trained with the same initial learning rate of $5 \times 10^{-5}$, which is the original setting of YOLOS for the neck-free detector. We do not change any hyperparameters used in transformer encoder and decoder for (Deformable) DETR; thus, the neck decoder of ViDT also consists of six deformable transformer layers using exactly the same hyperparameters. The only new hyperparameter introduced, the distillation coefficient $\lambda_{dis}$ in Eq. (2), is set to be 4. For fair comparison, knowledge distillation is not applied for ViDT in the main experiment in Section 4.1. The efficacy of knowledge distillation with token matching is verified independently in Section 4.3.2. Auxiliary decoding loss and iterative box refinement are applied to the compared methods if applicable.

Regarding the resolution of input images, we use scale augmentation that resizes them such that the shortest side is at least 480 and at most 800 pixels while the longest at most 1333 (Wu et al., 2019). More details of the experiment configuration can be found in Appendix B.3–B.5. All the source code and trained models will be made available to the public at https://github.com/naver-ai/vidt.

### 4.1 MAIN EXPERIMENTS WITH MICROSOFT COCO BENCHMARK

Table 2 compares ViDT with DETR (ViT) and YOLOS w.r.t their AP, FPS, # parameters, where the two variants of DETR (ViT) are simply named DETR and Deformable DETR. We report the result of ViDT without using knowledge distillation for fair comparison. A summary plot is provided in Figure 1. The experimental comparisons with CNN backbones are provided in Appendix C.1.

---

[3]Swin-nano is designed such that its channel dimension is half that of Swin-tiny. Please see Appendix B.1.

| Method | Backbone | Epochs | AP | AP$_{50}$ | AP$_{75}$ | AP$_S$ | AP$_M$ | AP$_L$ | Param. | FPS |
|---|---|---|---|---|---|---|---|---|---|---|
| DETR | DeiT-tiny | 50 | 30.0 | 49.2 | 30.5 | 9.9 | 30.8 | 50.6 | 24M | 10.9 (13.1) |
| | DeiT-small | 50 | 32.4 | 52.5 | 33.2 | 11.3 | 33.5 | 53.7 | 39M | 7.8 (8.8) |
| | DeiT-base | 50 | 37.1 | 59.2 | 38.4 | 14.7 | 39.4 | 52.9 | 0.1B | 4.3 (4.9) |
| | Swin-nano | 50 | 27.8 | 47.5 | 27.4 | 9.0 | 29.2 | 44.9 | 24M | 24.7 (46.1) |
| | Swin-tiny | 50 | 34.1 | 55.1 | 35.3 | 12.7 | 35.9 | 54.2 | 45M | 19.3 (28.1) |
| | Swin-small | 50 | 37.6 | 59.0 | 39.0 | 15.9 | 40.1 | 58.9 | 66M | 13.5 (17.7) |
| | Swin-base | 50 | 40.7 | 62.9 | 42.7 | 18.3 | 44.1 | 62.4 | 0.1B | 9.7 (12.6) |
| Deformable DETR | DeiT-tiny | 50 | 40.8 | 60.1 | 43.6 | 21.4 | 43.4 | 58.2 | 18M | 12.4 (16.3) |
| | DeiT-small | 50 | 43.6 | 63.7 | 46.5 | 23.3 | 47.1 | 62.1 | 35M | 8.5 (10.2) |
| | DeiT-base | 50 | 46.4 | 67.3 | 49.4 | 26.7 | 50.1 | 65.4 | 0.1B | 4.4 (5.3) |
| | Swin-nano | 50 | 43.1 | 61.4 | 46.3 | 25.9 | 45.2 | 59.4 | 18M | 7.0 (7.8) |
| | Swin-tiny | 50 | 47.0 | 66.8 | 50.8 | 28.1 | 49.8 | 63.9 | 39M | 6.3 (7.0) |
| | Swin-small | 50 | 49.0 | 68.9 | 52.9 | 30.3 | 52.8 | 66.6 | 60M | 5.5 (6.1) |
| | Swin-base | 50 | 51.4 | 71.7 | 56.2 | 34.5 | 55.1 | 67.5 | 0.1B | 4.8 (5.4) |
| YOLOS | DeiT-tiny | 150 | 30.4 | 48.6 | 31.1 | 12.4 | 31.8 | 48.2 | 6M | 28.1 (31.3) |
| | DeiT-small | 150 | 36.1 | 55.7 | 37.6 | 15.6 | 38.4 | 55.3 | 30M | 9.3 (11.8) |
| | DeiT-base | 150 | 42.0 | 62.2 | 44.5 | 19.5 | 45.3 | 62.1 | 0.1B | 3.9 (5.4) |
| ViDT (w.o. Neck) | Swin-nano | 150 | 28.7 | 48.6 | 28.5 | 12.3 | 30.7 | 44.1 | 7M | 36.5 (64.4) |
| | Swin-tiny | 150 | 36.3 | 56.3 | 37.8 | 16.4 | 39.0 | 54.3 | 29M | 28.6 (32.1) |
| | Swin-small | 150 | 41.6 | 62.7 | 43.9 | 20.1 | 45.4 | 59.8 | 52M | 16.8 (18.8) |
| | Swin-base | 150 | 43.2 | 64.2 | 45.9 | 21.9 | 46.9 | 63.2 | 91M | 11.5 (12.5) |
| ViDT | Swin-nano | 50 | 40.4 | 59.6 | 43.3 | 23.2 | 42.5 | 55.8 | 16M | 20.0 (45.8) |
| | Swin-tiny | 50 | 44.8 | 64.5 | 48.7 | 25.9 | 47.6 | 62.1 | 38M | 17.2 (26.5) |
| | Swin-small | 50 | 47.5 | 67.7 | 51.4 | 29.2 | 50.7 | 64.8 | 61M | 12.1 (16.5) |
| | Swin-base | 50 | 49.2 | 69.4 | 53.1 | 30.6 | 52.6 | 66.9 | 0.1B | 9.0 (11.6) |

Table 2. Comparison of ViDT with other compared detectors on COCO2017 val set. Two neck-free detectors, YOLOS and ViDT (w.o. Neck) are trained for 150 epochs due to the slow convergence. FPS is measured with batch size 1 of $800 \times 1333$ resolution on a single Tesla V100 GPU, where the value inside the parentheses is measured with batch size 4 of the same resolution to maximize GPU utilization.

**Highlights.** ViDT achieves the best trade-off between AP and FPS. With its high scalability, it performs well even for Swin-base of 0.1 billion parameters, which is 2x faster than Deformable DETR with similar AP. Besides, ViDT shows 40.4AP only with 16M parameters; it is 6.3–12.6AP higher than those of DETR (swin-nano) and DETR (swin-tiny), which exhibit similar FPS of 19.3–24.7.

**ViDT vs. Deformable DETR.** Thanks to the use of multi-scale features, Deformable DETR exhibits high detection performance in general. Nevertheless, its encoder and decoder structure in the neck becomes a critical bottleneck in computation. In particular, the encoder with multi-layer deformable transformers adds considerable overhead to transform multi-scale features by attention. Thus, it shows very low FPS although it achieves higher AP with a relatively small number of parameters. In contrast, ViDT removes the need for a transformer encoder in the neck by using Swin Transformer with RAM as its body, directly extracting multi-scale features suitable for object detection.

**ViDT (w.o. Neck) vs. YOLOS.** For the comparison with YOLOS, we train ViDT without using its neck component. These two neck-free detectors show relatively low AP compared with other detectors in general. In terms of speed, YOLOS exhibits much lower FPS than ViDT (w.o. Neck) because of its quadratic computational complexity for attention. However, ViDT (w.o. Neck) extends Swin Transformers with RAM, thus requiring linear complexity for attention. Hence, it shows AP comparable to YOLOS for various backbone size, but its FPS is much higher.

One might argue that better integration could be also achieved by (1) Deformable DETR without its neck encoder because its neck decoder also has [DET] $\times$ [PATCH] cross-attention, or (2) YOLOS with VIDT's neck decoder because of the use of multiple auxiliary techniques. Such integration is actually not effective; the former significantly drops AP, while the latter has a much greater drop in FPS than an increase in AP. The detailed analysis can be found in Appendix C.2.

## 4.2 Ablation Study on Reconfigured Attention Module (RAM)

We extend Swin Transformer with RAM to extract fine-grained features for object detection without maintaining an additional transformer encoder in the neck. We provide an ablation study on the two main considerations for RAM, which leads to high accuracy and speed. To reduce the influence of secondary factors, we mainly use our neck-free version, ViDT (w.o. Neck), for the ablation study.

### 4.2.1 SPATIAL POSITIONAL ENCODING

Spatial positional encoding is essential for [DET] × [PATCH] attention in RAM. Typically, the spatial encoding can be added to the [PATCH] tokens before or after the projection layer in Figure 3. We call the former "pre-addition" and the latter "post-addition". For each one, we can design the encoding in a sinusoidal or learnable manner (Carion et al.,

| Method | None | Pre-addition | | Post-addition | |
|--------|------|------|--------|------|--------|
| Type | None | Sin. | Learn. | Sin. | Learn. |
| AP | 23.7 | 28.7 | 27.4 | 28.0 | 24.1 |

Table 3. Results for different spatial encodings for [DET] × [PATCH] cross-attention.

2020). Table 3 contrasts the results with different spatial positional encodings with ViDT (w.o. Neck). Overall, pre-addition results in performance improvement higher than post-addition, and specifically, the sinusoidal encoding is better than the learnable one; thus, the 2D inductive bias of the sinusoidal spatial encoding is more helpful in object detection. In particular, pre-addition with the sinusoidal encoding increases AP by 5.0 compared to not using any encoding.

### 4.2.2 SELECTIVE [DET] × [PATCH] CROSS-ATTENTION

The addition of cross-attention to Swin Transformer inevitably entails computational overhead, particularly when the number of [PATCH] is large. To alleviate such overhead, we selectively enable cross-attention in RAM at the last stage of Swin Transformer; this is shown to greatly improve FPS, but barely drop AP. Table 4 summarizes AP and FPS when used different selective strategies for the cross-attention, where Swin Transformer consists of four stages in total. It is interesting that all the strategies exhibit similar AP as long as cross-attention is activated at the last stage. Since features are extracted in a bottom-up manner as they go through the stages, it seems difficult to directly obtain useful information about the target object at the low level of stages. Thus, only using the last stage is the best design choice in terms of high AP and FPS due to the smallest number of [PATCH] tokens.

Meanwhile, the detection fails completely or the performance significantly drops if all the stages are not involved due to the lack of interaction between [DET] and [PATCH] tokens that spatial positional encoding is associated with. A more detailed analysis of the [DET] × [PATCH] cross-attention and [DET] × [DET] self-attention is provided in appendices C.3 and C.4.

| Stage Ids | {1, 2, 3, 4} | | {2, 3, 4} | | {3, 4} | | {4} | | { } | |
|-----------|------|------|------|------|------|------|------|------|------|------|
| Metric | AP | FPS | AP | FPS | AP | FPS | AP | FPS | AP | FPS |
| w.o. Neck | 29.0 | 21.8 | 28.8 | 29.1 | 28.5 | 34.3 | 28.7 | 36.5 | FAIL | 37.7 |
| w. Neck | 40.3 | 14.6 | 40.1 | 18.0 | 40.3 | 19.5 | 40.4 | 20.0 | 37.1 | 20.5 |

Table 4. AP and FPS comparison with different selective cross-attention strategies.

## 4.3 ABLATION STUDY ON ADDITIONAL TECHNIQUES

We analyze the performance improvement of two additional techniques, namely auxiliary decoding loss and iterative box refinement, and the proposed distillation approach in Section 3.3. Furthermore, we introduce a simple technique that can expedite the inference speed of ViDT by dropping unnecessary decoding layers at inference time.

### 4.3.1 AUXILIARY DECODING LOSS AND ITERATIVE BOX REFINEMENT

To thoroughly verify the efficacy of auxiliary decoding loss and iterative box refinement, we extend them even for the neck-free detector like YOLOS; the principle of them is applied to the encoding layers in the body, as opposed to the conventional way of using the decoding layers in the neck. Table 5 shows the performance of the two neck-free detectors, YOLOS and ViDT (w.o. Neck), decreases considerably with the two techniques. The use of them in the encoding layers is likely to negatively affect feature extraction of the transformer encoder. In contrast, an opposite trend is observed with the neck component. Since

| | Aux. $\ell$ | Box Ref. | Neck | AP | $\Delta$ |
|--------|------|------|------|------|------|
| YOLOS | | | | 30.4 | |
| | ✓ | | | 29.2 | −1.2 |
| | ✓ | ✓ | | 20.1 | −10.3 |
| ViDT | | | | 28.7 | |
| | ✓ | | | 27.2 | −1.6 |
| | ✓ | ✓ | | 22.9 | −5.9 |
| | ✓ | | ✓ | 36.2 | +7.4 |
| | ✓ | ✓ | ✓ | 40.4 | +11.6 |

Table 5. Effect of extra techniques with YOLOS (DeiT-tiny) and ViDT (Swin-nano).

the neck decoder is decoupled with the feature extraction in the body, the two techniques make a synergistic effect and thus show significant improvement in AP. These results justify the use of the neck decoder in ViDT to boost object detection performance.

### 4.3.2 KNOWLEDGE DISTILLATION WITH TOKEN MATCHING

We show that a small ViDT model can benefit from a large ViDT model via knowledge distillation. The proposed token matching is a new concept of knowledge distillation for object detection, especially for a fully transformer-based object detector. Compared to very complex distillation methods that rely on heuristic rules with multiple hyperparameters (Chen et al., 2017; Dai et al., 2021), it simply matches some tokens with

| Student | ViDT (Swin-nano) | | ViDT (Swin-tiny) | |
|---|---|---|---|---|
| Teacher | ViDT (small) | ViDT (base) | ViDT (small) | ViDT (base) |
| $\lambda_{dis} = 0$ | 40.4 | | 44.8 | |
| $\lambda_{dis} = 2$ | 41.4 | 41.4 | 45.6 | 46.1 |
| $\lambda_{dis} = 4$ | 41.5 | 41.9 | 45.8 | 46.5 |

Table 6. AP comparison of student models associated with different teacher models.

a single hyperparameter, the distillation coefficient $\lambda_{dis}$. Table 6 summarizes the AP improvement via knowledge distillation with token matching with varying distillation coefficients. Overall, the larger the size of the teacher model, the greater gain to the student model. Regarding coefficients, in general, larger values achieve better performance. Distillation increases AP by 1.0–1.7 without affecting the inference speed of the student model.

### 4.3.3 DECODING LAYER DROP

ViDT has six layers of transformers as its neck decoder. We emphasize that not all layers of the decoder are required at inference time for high performance. Table 7 show the performance of ViDT when dropping its decoding layer one by one from the top in the inference step. Although there is a trade-off relationship between accuracy and speed as the layers are detached from the model, there is no significant AP drop even when the two layers are removed. This technique is not designed for performance evaluation in Table 2

| Model | ViDT (Swin-nano) | | | ViDT (Swin-tiny) | | |
|---|---|---|---|---|---|---|
| Metric | AP | Param. | FPS | AP | Param. | FPS |
| 0 Drop | 40.4 | 16M | 20.0 | 44.8 | 38M | 17.2 |
| 1 Drop | 40.2 | 14M | 20.9 | 44.8 | 37M | 18.5 |
| 2 Drop | 40.0 | 13M | 22.3 | 44.5 | 35M | 19.6 |
| 3 Drop | 38.6 | 12M | 24.7 | 43.6 | 34M | 21.0 |
| 4 Drop | 36.8 | 11M | 26.0 | 41.9 | 33M | 22.4 |
| 5 Drop | 32.5 | 10M | 28.7 | 38.0 | 32M | 24.4 |

Table 7. Performance trade-off by decoding layer drop regarding AP, Param, and FPS.

with other methods, but we can accelerate the inference speed of a trained ViDT model to over 10% by dropping its two decoding layers without a much decrease in AP.

### 4.4 COMPLETE COMPONENT ANALYSIS

In this section, we combine all the proposed components (even with distillation and decoding layer drop) to achieve high accuracy and speed for object detection. As summarized in Table 8, there are four components: (1) RAM to extend Swin Transformer as a standalone object detector, (2) the neck decoder to exploit multi-scale features with two auxiliary techniques, (3) knowledge distillation to benefit from a large model, and (4) decoding layer drop to further accelerate inference speed. The performance of the final version is very outstanding; it achieves 41.7AP with reasonable FPS by only using 13M parameters when used Swin-nano as its backbone. Further, it only loses 2.7 FPS while exhibiting 46.4AP when used Swin-tiny. This indicates that a fully transformer-based object detector has the potential to be used as a generic object detector when further developed in the future.

| # | Component | | | | Swin-nano | | | | | Swin-tiny | | | | |
|---|---|---|---|---|---|---|---|---|---|---|---|---|---|---|
| | RAM | Neck | Distil | Drop | AP | $AP_{50}$ | $AP_{75}$ | Param. | FPS | AP | $AP_{50}$ | $AP_{75}$ | Param. | FPS |
| (1) | ✓ | | | | 28.7 | 48.6 | 28.5 | 7M | 36.5 | 36.3 | 56.3 | 37.8 | 29M | 28.6 |
| (2) | ✓ | ✓ | | | 40.4 | 59.6 | 43.3 | 16M | 20.0 | 44.8 | 64.5 | 48.7 | 38M | 17.2 |
| (3) | ✓ | ✓ | ✓ | | 41.9 | 61.1 | 45.0 | 16M | 20.0 | 46.5 | 66.3 | 50.2 | 38M | 17.2 |
| (4) | ✓ | ✓ | ✓ | ✓ | 41.7 | 61.0 | 44.8 | 13M | 22.3 | 46.4 | 66.3 | 50.2 | 35M | 19.6 |

Table 8. Detailed component analysis with Swin-nano and Swin-tiny.

## 5 CONCLUSION

We have explored the integration of vision and detection transformers to build an effective and efficient object detector. The proposed ViDT significantly improves the scalability and flexibility of transformer models to achieve high accuracy and inference speed. The computational complexity of its attention modules is linear w.r.t. image size, and ViDT synergizes several essential techniques to boost the detection performance. On the Microsoft COCO benchmark, ViDT achieves 49.2AP with a large Swin-base backbone, and 41.7AP with the smallest Swin-nano backbone and only 13M parameters, suggesting the benefits of using transformers for complex computer vision tasks.

## ETHICS STATEMENT

This paper deals with the topic of general object detection in computer vision. We propose a novel integration of vision and detection transformers for a fully transformer-based object detector. Therefore, we do not expect any potential negative social impact of our work.

## REPRODUCIBILITY STATEMENT

For reproducibility, we provide a detailed description of our experiment and hyperparameter settings in Appendix B. It includes the Swin-nano architecture (Appendix B.1), the pipelines of all compared object detectors (Appendix B.2), hyperparameters of neck transformers (Appendix B.3), detailed implementation (Appendix B.4), and training configuration (Appendix B.5). We will release the code and trained models upon acceptance.

## ACKNOWLEDGMENTS

We thank NAVER AI Lab members for valuable discussion and advice. NAVER Smart Machine Learning (NSML) (Kim et al., 2018) has been used for experiment. M.-H. Yang is supported in part by the NSF CAREER grant 1149783.

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

# An Efficient and Effective Fully Transformer-based Object Detector (Supplementary Material)

## A    RECONFIGURED ATTENTION MODULE

The proposed RAM in Figure 3 performs three attention operations, namely $[\texttt{PATCH}] \times [\texttt{PATCH}]$, $[\texttt{DET}] \times [\texttt{PATCH}]$, and $[\texttt{DET}] \times [\texttt{DET}]$ attention. This section provides (1) computational complexity analysis and (2) further algorithmic design for $[\texttt{DET}]$ tokens.

### A.1    COMPUTATIONAL COMPLEXITY ANALYSIS

We analyze the computational complexity of the proposed RAM compared with the attention used in YOLOS. The analysis is based on the computational complexity of basic building blocks for Canonical and Swin Transformer, which is summarized in Table 9[4], where $\mathsf{T}_1$ and $\mathsf{T}_2$ is the number of tokens for self- and cross-attention, and $d$ is the embedding dimension.

| Transformer | Canonical Transformer | | Swin Transformer |
|---|---|---|---|
| Attention | Global Self-attention | Global Cross-attention | Local Self-attention |
| Complexity | $\mathcal{O}(d^2\mathsf{T}_1 + d\mathsf{T}_1^2)$ | $\mathcal{O}(d^2(\mathsf{T}_1 + \mathsf{T}_2) + d\mathsf{T}_1\mathsf{T}_2)$ | $\mathcal{O}(d^2\mathsf{T}_1 + dk^2\mathsf{T}_1)$ |

Table 9. Computational complexity of attention modules: In the canonical transformer, the complexity of global self-attention is $\mathcal{O}(d^2\mathsf{T}_1 + d\mathsf{T}_1^2)$, where $\mathcal{O}(d^2\mathsf{T}_1)$ is the cost of computing the query, key, and value embeddings and $\mathcal{O}(d\mathsf{T}_1^2)$ is the cost of computing the attention weights. The complexity of global cross-attention is $\mathcal{O}(d^2(\mathsf{T}_1 + \mathsf{T}_2) + d\mathsf{T}_1\mathsf{T}_2)$, which is the interaction between the two different tokens $\mathsf{T}_1$ and $\mathsf{T}_2$. In contrast, Swin Transformer achieves much lower attention complexity of $\mathcal{O}(d^2\mathsf{T}_1 + dk^2\mathsf{T}_1)$ with window partitioning, where $k$ is the width and height of the window ($k << \mathsf{T}_1, \mathsf{T}_2$).

Let $\mathsf{P}$ and $\mathsf{D}$ be the number of $[\texttt{PATCH}]$ and $[\texttt{DET}]$ tokens ($\mathsf{D} << \mathsf{P}$ in practice, e.g., $\mathsf{P} = 66,650$ and $\mathsf{D} = 100$ at the first stage of ViDT). Then, the computational complexity of the attention module for YOLOS and ViDT (RAM) is derived as below, also summarized in Table 10:

- **YOLOS Attention**: $[\texttt{DET}]$ tokens are simply appended to $[\texttt{PATCH}]$ tokens to perform global self-attention on $[\texttt{PATCH}, \texttt{DET}]$ tokens (i.e., $\mathsf{T}_1 = \mathsf{P} + \mathsf{D}$). Thus, the computational complexity is $\mathcal{O}(d^2(\mathsf{P} + \mathsf{D}) + d(\mathsf{P} + \mathsf{D})^2)$, which is *quadratic* to the number of $[\texttt{PATCH}]$ tokens. If breaking down the total complexity, we obtain $\mathcal{O}\big((d^2\mathsf{P} + d\mathsf{P}^2) + (d^2\mathsf{D} + d\mathsf{D}^2) + d\mathsf{P}\mathsf{D}\big)$, where the first and second terms are for the global self-attention for $[\texttt{PATCH}]$ and $[\texttt{DET}]$ tokens, respectively, and the last term is for the global cross-attention between them.

- **ViDT (RAM) Attention**: RAM performs the three different attention operations: (1) $[\texttt{PATCH}] \times [\texttt{PATCH}]$ local self-attention with window partition, $\mathcal{O}(d^2\mathsf{P} + dk^2\mathsf{P})$; (2) $[\texttt{DET}] \times [\texttt{DET}]$ global self-attention, $\mathcal{O}(d^2\mathsf{D} + d\mathsf{D}^2)$; (3) $[\texttt{DET}] \times [\texttt{PATCH}]$ global cross-attention, $\mathcal{O}(d^2(\mathsf{D} + \mathsf{P}) + d\mathsf{D}\mathsf{P})$. In total, the computational complexity of RAM is $\mathcal{O}(d^2(\mathsf{D} + \mathsf{P}) + dk^2\mathsf{P} + d\mathsf{D}^2 + d\mathsf{D}\mathsf{P})$, which is *linear* to the number of $[\texttt{PATCH}]$ tokens.

Consequently, the complexity of RAM is much lower than the attention module used in YOLOS since $\mathsf{D} << \mathsf{P}$. Note that only RAM achieves the *linear* complexity to the patch tokens. In addition, one might argue that YOLOS can be efficient if the cross-attention is selectively removed similar to RAM. Even if we remove the complexity $\mathcal{O}(d\mathsf{P}\mathsf{D})$ for the global cross-attention, the computational complexity is $\mathcal{O}(d^2(\mathsf{P} + \mathsf{D}) + d\mathsf{P}^2 + d\mathsf{D}^2)$, which is still *quadratic* to the number of $[\texttt{PATCH}]$ tokens.

| Attention Type | YOLOS | ViDT |
|---|---|---|
| $[\texttt{PATCH}] \times [\texttt{PATCH}]$ | $\mathcal{O}(d^2\mathsf{P} + d\mathsf{P}^2)$ | $\mathcal{O}(d^2\mathsf{P} + dk^2\mathsf{P})$ |
| $[\texttt{DET}] \times [\texttt{DET}]$ | $\mathcal{O}(d^2\mathsf{D} + d\mathsf{D}^2)$ | $\mathcal{O}(d^2\mathsf{D} + d\mathsf{D}^2)$ |
| $[\texttt{DET}] \times [\texttt{PATCH}]$ | $\mathcal{O}(d\mathsf{P}\mathsf{D})$ | $\mathcal{O}(d^2(\mathsf{D} + \mathsf{P}) + d\mathsf{D}\mathsf{P})$ |
| Total Complexity | $\mathcal{O}(d^2(\mathsf{P} + \mathsf{D}) + d(\mathsf{P} + \mathsf{D})^2)$ | $\mathcal{O}(d^2(\mathsf{D} + \mathsf{P}) + dk^2\mathsf{P} + d\mathsf{D}^2 + d\mathsf{D}\mathsf{P})$ |

Table 10. Summary of computational complexity for different attention operations used in YOLOS and ViDT (RAM), where $\mathsf{P}$ and $\mathsf{D}$ are the number of $[\texttt{PATCH}]$ and $[\texttt{DET}]$ tokens, respectively ($\mathsf{D} << \mathsf{P}$).

---

[4]We used the computational complexity reported in the original paper (Vaswani et al., 2017; Liu et al., 2021)

## A.2 ALGORITHMIC DESIGN FOR [DET] TOKENS

### A.2.1 BINDING [DET] × [DET] AND [DET] × [PATCH] ATTENTION

Binding the two attention modules is very simple in implementation. [DET] × [DET] and [DET] × [PATCH] attention is generating a new [DET] token, which aggregates relevant contents in [DET] and [PATCH] tokens, respectively. Since the two attention share exactly the same [DET] query embedding obtained after the projection as shown in Figure 3, they can be processed at once by performing the scaled-dot product between $[\text{DET}]_Q$ and $\big[[\text{DET}]_K, [\text{PATCH}]_K\big]$ embeddings, where $Q$, $K$ are the key and query, and $[\cdot]$ is the concatenation. Then, the obtained attention map is applied to the $\big[[\text{DET}]_V, [\text{PATCH}]_V\big]$ embeddings, where $V$ is the value and $d$ is the embedding dimension,

$$[\text{DET}]_{new} = \text{Softmax}\Big(\frac{[\text{DET}]_Q\big[[\text{DET}]_K, [\text{PATCH}]_K\big]^\top}{\sqrt{d}}\Big)\big[[\text{DET}]_V, [\text{PATCH}]_V\big]. \qquad (3)$$

This approach is commonly used in the recent Transformer-based architectures, such as YOLOS.

### A.2.2 EMBEDDING DIMENSION OF [DET] TOKENS

[DET] × [DET] attention is performed across all the stages, and the embedding dimension of [DET] tokens increases gradually like [PATCH] tokens. For the [PATCH] token, its embedding dimension is increased by concatenating nearby [PATCH] tokens in a grid. However, this mechanism is not applicable for [DET] tokens since we maintain the same number of [DET] tokens for detecting a fixed number of objects in a scene. Hence, we simply repeat a [DET] token multiple times along the embedding dimension to increase its size. This allows [DET] tokens to reuse all the projection and normalization layers in Swin Transformer without any modification.

## B EXPERIMENTAL DETAILS

### B.1 SWIN-NANO ARCHITECTURE

Due to the absence of Swin models comparable to Deit-tiny, we configure Swin-nano, which is a $0.25\times$ model of Swin-tiny such that it has 6M training parameters comparable to Deit-tiny. Table 11 summarizes the configuration of Swin Transformer models available, including the newly introduced Swinnano; S1–S4 indicates the four stages in Swin Transformer. The performance of all the pre-trained Swin Transformer models are summarized in Table 1 in the manuscript.

| Model | Channel | Layer Numbers | | | |
|---|---|---|---|---|---|
| Name | Dim. | S1 | S2 | S3 | S4 |
| Swin-nano | 48 | 2 | 2 | 6 | 2 |
| Swin-tiny | 96 | 2 | 2 | 6 | 2 |
| Swin-small | 128 | 2 | 2 | 18 | 2 |
| Swin-base | 192 | 2 | 2 | 18 | 2 |

Table 11. Swin Transformer Architecture.

### B.2 DETECTION PIPELINES OF ALL COMPARED DETECTORS

All the compared fully transformer-based detectors are composed of either (1) *body–neck–head* or (2) *body–head* structure, as summarized in Table 12. The main difference of ViDT is the use of reconfigured attention modules (RAM) for Swin Transformer, allowing the extraction of fine-grained detection features directly from the input image. Thus, Swin Transformer is extended to a standalone object detector called ViDT (w.o. Neck). Further, its extension to ViDT allows to use multi-scale features and multiple essential techniques for better detection, such as auxiliary decoding loss and iterative box refinement, by only maintaining a transformer decoder at the neck. Except for the two neck-free detector, YOLOS and ViDT (w.o. Neck), all the pipelines maintain multiple FFNs; that is, a single FFNs for each decoding layer at the neck for box regression and classification.

We believe that our proposed RAM can be combined with even other latest efficient vision transformer architectures, such as PiT (Heo et al., 2021), PVT (Wang et al., 2021) and Cross-ViT (Chen et al., 2021). We leave this as future work.

### B.3 HYPERPARAMETERS OF NECK TRANSFORMERS

The transformer decoder at the neck in ViDT introduces multiple hyperparameters. We follow exactly the same setting used in Deformable DETR. Specifically, we use six layers of deformable

| Pipeline | Body | Neck | | Head |
|---|---|---|---|---|
| Method Name | Feature Extractor | Tran. Encoder | Tran. Decoder | Prediction |
| DETR (DeiT) | DeiT Transformer | ◯ | ◯ | Multiple FFNs |
| DETR (Swin) | Swin Transformer | ◯ | ◯ | Multiple FFNs |
| Deformable DETR (DeiT) | DeiT Transformer | ◯† | ◯† | Multiple FFNs |
| Deformable DETR (Swin) | Swin Transformer | ◯† | ◯† | Multiple FFNs |
| YOLOS | DeiT Transformer | ✕ | ✕ | Single FFNs |
| ViDT (w.o. Neck) | Swin Transformer+RAM | ✕ | ✕ | Single FFNs |
| ViDT | Swin Transformer+RAM | ✕ | ◯† | Multiple FFNs |

Table 12. Comparison of detection pipelines for all available fully transformer-based object detectors, where † indicates that multi-scale deformable attention is used for neck transformers.

transformers with width 256; thus, the channel dimension of the [PATCH] and [DET] tokens extracted from Swin Transformer are reduced to 256 to be utilized as compact inputs to the decoder transformer. For each transformer layer, multi-head attention with eight heads is applied, followed by the point-wise FFNs of 1024 hidden units. Furthermore, an additive dropout of 0.1 is applied before the layer normalization. All the weights in the decoder are initialized with Xavier initialization. For (Deformable) DETR, the tranformer decoder receives a fixed number of learnable detection tokens. We set the number of detection tokens to 100, which is the same number used for YOLOS and ViDT.

## B.4 IMPLEMENTATION

### B.4.1 DETECTION HEAD FOR PREDICTION

The last [DET] tokens produced by the body or neck are fed to a 3-layer FFNs for bounding box regression and linear projection for classification,

$$\hat{B} = \text{FFN}_{\text{3-layer}}([\text{DET}]) \ \text{ and } \ \hat{P} = \text{Linear}([\text{DET}]). \tag{4}$$

For box regression, the FFNs produce the bounding box coordinates for $d$ objects, $\hat{B} \in [0, 1]^{d \times 4}$, that encodes the normalized box center coordinates along with its width and height. For classification, the linear projection uses a softmax function to produce the classification probabilities for all possible classes including the background class, $\hat{P} \in [0, 1]^{d \times (c+1)}$, where $c$ is the number of object classes. When deformable attention is used on the neck in Table 12, only $c$ classes are considered without the background class for classification. This is the original setting used in DETR, YOLOS (Carion et al., 2020; Fang et al., 2021) and Deformable DETR (Zhu et al., 2021).

### B.4.2 LOSS FUNCTION FOR TRAINING

All the methods adopts the loss function of (Deformable) DETR. Since the detection head return a fixed-size set of $d$ bounding boxes, where $d$ is usually larger than the number of actual objects in an image, Hungarian matching is used to find a bipartite matching between the predicted box $\hat{B}$ and the ground-truth box $B$. In total, there are three types of training loss: a classification loss $\ell_{cl}$[5], a box distance $\ell_{l_1}$, and a GIoU loss $\ell_{iou}$ (Rezatofighi et al., 2019),

$$\ell_{cl}(i) = -\log \hat{P}_{\sigma(i),c_i}, \ \ \ell_{\ell_1}(i) = ||B_i - \hat{B}_{\sigma(i)}||_1, \text{ and}$$
$$\ell_{iou}(i) = 1 - \Big( \frac{|B_i \cap \hat{B}_{\sigma(i)}|}{|B_i \cup \hat{B}_{\sigma(i)}|} - \frac{|\mathsf{B}(B_i, \hat{B}_{\sigma(i)}) \backslash B_i \cup \hat{B}_{\sigma(i)}|}{|\mathsf{B}(B_i, \hat{B}_{\sigma(i)})|} \Big), \tag{5}$$

where $c_i$ and $\sigma(i)$ are the target class label and bipartite assignment of the $i$-th ground-truth box, and $\mathsf{B}$ returns the largest box containing two given boxes. Thus, the final loss of object detection is a linear combination of the three types of training loss,

$$\ell = \lambda_{cl} \ell_{cl} + \lambda_{\ell_1} \ell_{\ell_1} + \lambda_{iou} \ell_{iou}. \tag{6}$$

---

[5]Cross-entropy loss is used with standard transformer architectures, while focal loss (Lin et al., 2017) is used with deformable transformer architecture.

| Method | Backbone | Epochs | AP | AP$_{50}$ | AP$_{75}$ | AP$_S$ | AP$_M$ | AP$_L$ | Param. | FPS |
|---|---|---|---|---|---|---|---|---|---|---|
| DETR | ResNet-50 | 500 | 42.0 | 62.4 | 44.2 | 20.5 | 45.8 | 61.1 | 41M | 22.8 (38.6) |
| DETR-DC5 | ResNet-50 | 500 | 43.3 | 63.1 | 45.9 | 22.5 | 47.3 | 61.1 | 41M | 12.8 (14.2) |
| DETR-DC5 | ResNet-50 | 50 | 35.3 | 55.7 | 36.8 | 15.2 | 37.5 | 53.6 | 41M | 12.8 (14.2) |
| Deform. DETR | ResNet-50 | 50 | 45.4 | 64.7 | 49.0 | 26.8 | 48.3 | 61.7 | 40M | 13.7 (19.4) |
| ViDT | Swin-tiny | 50 | 44.8 | 64.5 | 48.7 | 25.9 | 47.6 | 62.1 | 38M | 17.2 (26.5) |
| ViDT | Swin-tiny | 150 | 47.2 | 66.7 | 51.4 | 28.4 | 50.2 | 64.7 | 38M | 17.2 (26.5) |

Table 13. Evaluations of ViDT with other detectors using CNN backbones on COCO2017 val set. FPS is measured with batch size 1 of $800 \times 1333$ resolution on a single Tesla V100 GPU, where the value inside the parentheses is measured with batch size 4 of the same resolution to maximize GPU utilization.

| Method | Backbone | AP | AP$_{50}$ | AP$_{75}$ | AP$_S$ | AP$_M$ | AP$_L$ | Param. | FPS |
|---|---|---|---|---|---|---|---|---|---|
| Deformable DETR | Swin-nano | 43.1 | 61.4 | 46.3 | 25.9 | 45.2 | 59.4 | 17M | 7.0 |
| − neck encoder | | 34.0 | 52.8 | 35.6 | 18.0 | 36.3 | 48.4 | 14M | 22.4 |
| YOLOS | DeiT-tiny | 30.4 | 48.6 | 31.1 | 12.4 | 31.8 | 48.2 | 6M | 28.1 |
| + neck decoder | | 38.1 | 57.1 | 40.2 | 20.1 | 40.2 | 56.0 | 14M | 17.1 |
| ViDT | Swin-nano | 40.4 | 59.6 | 43.3 | 23.2 | 42.5 | 55.8 | 16M | 20.0 |
| + neck encoder | | 46.1 | 64.1 | 49.7 | 28.5 | 48.7 | 61.7 | 19M | 6.3 |

Table 14. Variations of Deformable DETR, YOLOS, and ViDT with respect to their neck structure. They are trained for 50 epochs with the same configuration used in our main experimental results.

The coefficient for each training loss is set to be $\lambda_{cl} = 1$, $\lambda_{\ell_1} = 5$, and $\lambda_{iou} = 2$. If we leverage auxiliary decoding loss, the final loss is computed for every detection head separately and merged with equal importance. Additionally, ViDT adds the distillation loss in Eq. (2) to the final loss if the distillation approach in Section 3.3 is enabled for training.

### B.5 TRAINING CONFIGURATION

We train ViDT for 50 epochs using AdamW (Loshchilov & Hutter, 2019) with the same initial learning rate of $10^{-4}$ for its body, neck and head. The learning rate is decayed by cosine annealing with batch size of 16, weight decay of $1 \times 10^{-4}$, and gradient clipping of 0.1. In contrast, ViDT (w.o. Neck) is trained for 150 epochs using AdamW with the initial learning rate of $5 \times 10^{-5}$ by cosine annealing. The remaining configuration is the same as for ViDT.

Regarding DETR (ViT), we follow the setting of Deformable DETR. Thus, all the variants of this pipeline are trained for 50 epochs with the initial learning rate of $10^{-5}$ for its pre-trained body (ViT backbone) and $10^{-4}$ for its neck and head. Their learning rates are decayed at the 40-th epoch by a factor of 0.1. Meanwhile, the results of YOLOS are borrowed from the original paper (Fang et al., 2021) except YOLOS (DeiT-tiny); since the result of YOLOS (DeiT-tiny) for $800 \times 1333$ is not reported in the paper, we train it by following the training configuration suggested by authors.

## C SUPPLEMENTARY EVALUATION

### C.1 COMPARISON WITH OBJECT DETECTOR USING CNN BACKBONE

We compare ViDT with (Deformable) DETR using the ResNet-50 backbone, as summarized in Table 13, where all the results except ViDT are borrowed from (Carion et al., 2020; Zhu et al., 2021), and DETR-DC5 is a modification of DETR to use a dilated convolution at the last stage in ResNet. For a fair comparison, we compare ViDT (Swin-tiny) with similar parameter numbers. In general, ViDT shows a better trade-off between AP and FPS even compared with (Deformable) DETR with the ResNet-50. Specifically, ViDT achieves FPS much higher than DETR-DC5 and Deformable DETR with competitive AP. Particularly when training ViDT for 150 epochs, ViDT outperforms other compared methods using the ResNet-50 backbone in terms of both AP and FPS.

### C.2 VARIATIONS OF EXISTING PIPELINES

We study more variations of existing detection methods by modifying their original pipelines in Table 12. Thus, we remove the neck encoder of Deformable DETR to increase its efficiency, while adding a neck decoder to YOLOS to leverage multi-scale features along with auxiliary decoding

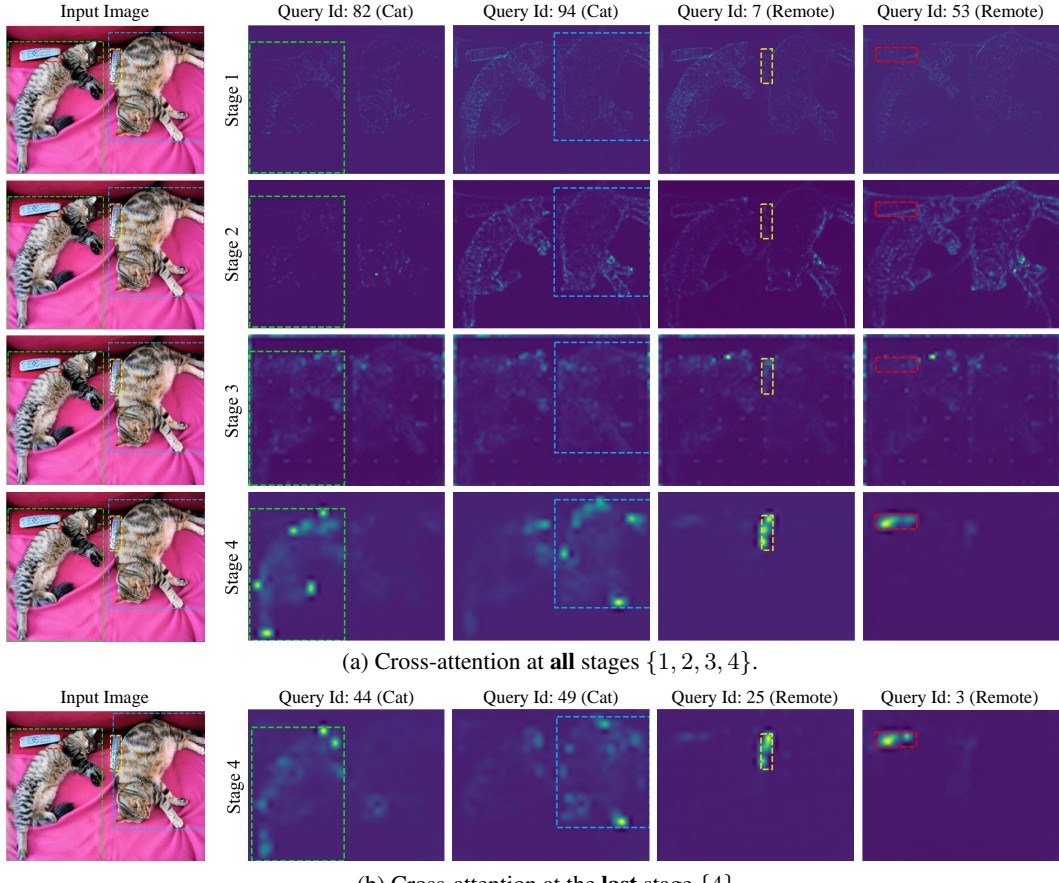

(a) Cross-attention at **all** stages $\{1, 2, 3, 4\}$.

(b) Cross-attention at the **last** stage $\{4\}$.

Figure 4. Visualization of the attention map for cross-attention with ViDT (Swin-nano).

loss and iterative box refinement. Note that these modified versions follow exactly the same detection pipeline with ViDT, maintaining a *encoder-free* neck between their body and head. Table 14 summarizes the performance of all the variations in terms of AP, FPS, and the number of parameters.

**Deformable DETR** shows significant improvement in FPS $(+14.4)$ but its AP drops sharply $(-9.1)$ when its neck encoder is removed. Thus, it is difficult to obtain fine-grained object detection representation directly from the raw ViT backbone without using an additional neck encoder. However, ViDT compensates for the effect of the neck encoder by adding $\lceil$DET$\rceil$ tokens into the body (backbone), thus successfully removing the computational bottleneck without compromising AP; it maintains 6.4 higher AP compared with the neck encoder-free Deformable DETR (the second row) while achieving similar FPS. This can be attributed to that RAM has a great contribution to the performance w.r.t AP and FPS, especially for the trade-off between them.

**YOLOS** shows a significant gain in AP $(+7.7)$ while losing FPS $(-11.0)$ when the neck decoder is added. Unlike Deformable DETR, its AP significantly increases even without the neck encoder due to the use of a standalone object detector as its backbone (i.e., the modified DeiT in Figure 2(b)). However, its AP is lower than ViDT by 2.3AP. Even worse, it is not scalable for large models because of its quadratic computational cost for attention. Therefore, in the aspects of accuracy and speed, ViDT maintains its dominance compared with the two carefully tuned baselines.

For a complete analysis, we additionally add a neck encoder to ViDT. The inference speed of ViDT degrades drastically by 13.7 because of the self-attention for multi-scale features at the neck encoder. However, it is interesting to see the improvement of AP by 5.7 while adding only 3M parameters; it is 3.0 higher even than Deformable DETR. This indicates that lowering the computational complexity of the encoder and thus increasing its utilization could be another possible direction for a fully transformer-based object detector.

## C.3 [DET] × [PATCH] ATTENTION IN RAM

In Section 4.2.2, it turns out that the cross-attention in RAM is only necessary at the last stage of Swin Transformer; all the different selective strategies show similar AP as long as cross-attention is activated at the last stage. Hence, we analyze the attention map obtained by the cross-attention in RAM. Figure 4 shows attention maps for the stages of Swin Transformer where cross-attention is utilized; it contrasts (a) ViDT with cross-attention at all stages and (b) ViDT with cross-attention at the last stage. Regardless of the use of cross-attention at the lower stage, it is noteworthy that the finally obtained attention map at the last stage is almost the same. In particular, the attention map at Stage 1–3 does not properly focus the features on the target object, which is framed by the bounding box. In addition, the attention weights (color intensity) at Stage 1–3 are much lower than those at Stage 4. Since features are extracted from a low level to a high level in a bottom-up manner as they go through the stages, it seems difficult to directly get information about the target object with such low-level features at the lower level of stages. Therefore, this analysis provides strong empirical evidence for the use of selective [DET] × [PATH] cross-attention.

## C.4 [DET] × [DET] ATTENTION IN RAM

Another possible consideration for ViDT is the use of [DET] × [DET] self-attention in RAM. We conduct an ablation study by removing the [DET] × [DET] attention one by one from the bottom stage, and summarize the results in Table 15. When all the [DET] × [DET] self-attention are removed, (5) the AP drops by 0.7, which is a meaningful performance degradation. On the other hand, as long as the self-attention is activated at the last two stages, (1) – (3) all the strategies exhibit similar AP. Therefore, only

|  | | Stage Id | | | | Swin-nano | |
| --- | --- | --- | --- | --- | --- | --- | --- |
| # | 1 | 2 | 3 | 4 | | AP | FPS |
| (1) | ✓ | ✓ | ✓ | ✓ | | 40.4 | 20.0 |
| (2) | | ✓ | ✓ | ✓ | | 40.3 | 20.1 |
| (3) | | | ✓ | ✓ | | 40.4 | 20.2 |
| (4) | | | | ✓ | | 40.1 | 20.3 |
| (5) | | | | | | 39.7 | 20.4 |

Table 15. AP and FPS comparison with different [DET] × [DET] self-attention strategies with ViDT.

keeping [DET] × [DET] self-attention at the last two stages can further increase FPS (+0.2) without degradation in AP. This observation could be used as another design choice for the AP and FPS trade-off. Therefore, we believe that [DET] × [DET] self-attention is meaningful to use in RAM.

# D PRELIMINARIES: TRANSFORMERS

A transformer is a deep model that entirely relies on the self-attention mechanism for machine translation (Vaswani et al., 2017). In this section, we briefly revisit the standard form of the transformer.

**Single-head Attention.** The basic building block of the transformer is a self-attention module, which generates a weighted sum of the values (contents), where the weight assigned to each value is the attention score computed by the scaled dot-product between its query and key. Let $W_Q$, $W_K$, and $W_V$ be the learned projection matrices of the attention module, and then the output is generated by

$$\text{Attention}(Z) = \text{softmax}\big(\frac{(ZW_Q)(ZW_K)^\top}{\sqrt{d}}\big)(ZW_V) \in \mathbb{R}^{hw \times d},$$

$$\text{where } W_Q, W_K, W_V \in \mathbb{R}^{d \times d}. \tag{7}$$

**Multi-head Attention.** It is beneficial to maintain multiple heads such that they repeat the linear projection process $k$ times with different learned projection matrices. Let $W_{Q_i}$, $W_{K_i}$, and $W_{V_i}$ be the learned projection matrices of the $i$-th attention head. Then, the output is generated by the concatenation of the results from all heads,

$$\text{Multi-Head}(Z) = [\text{Attention}_1(Z), \text{Attention}_2(Z), \ldots, \text{Attention}_k(Z)] \in \mathbb{R}^{hw \times d},$$

$$\text{where } \forall_i W_{Q_i}, W_{K_i}, W_{V_i} \in \mathbb{R}^{d \times (d/k)}. \tag{8}$$

Typically, the dimension of each head is divided by the total number of heads.

**Feed-Forward Networks (FFNs).** The output of the multi-head attention is fed to the point-wise FFNs, which performs the linear transformation for each position separately and identically to allow the model focusing on the contents of different representation subspaces. Here, the residual connec-

tion and layer normalization are applied before and after the FFNs. The final output is generated by

$$H = \text{LayerNorm}(\text{Dropout}(H') + H''),$$

where $H' = \text{FFN}(H'')$ and $H'' = \text{LayerNorm}(\text{Dropout}(\text{Multi-Head}(Z)) + Z).$ \hfill (9)

**Multi-Layer Transformers.** The output of a previous layer is fed directly to the input of the next layer. Regarding the positional encoding, the same value is added to the input of each attention module for all layers.

