# OpenReview forum: "ViDT: An Efficient and Effective Fully Transformer-based Object Detector"
_ICLR.cc/2022/Conference — ICLR 2022 Poster_

### Official Review · Reviewer_JKxV · 2021-10-31

**Correctness:** 4
**Technical Novelty And Significance:** 2
**Empirical Novelty And Significance:** 4
**Recommendation:** 8
**Confidence:** 5

**Main Review:**

#### Strengths

- This paper is well motivated and clearly written. It proposes a high-performance Detection Transformer coined as ViDT, which synergizes the best of Swin Transformer, YOLOS, as well as DETR (Deformable DETR) architectural design and for the first time demonstrates that the Detection Transformer family can achieve a very competitive accuracy-speed trade-off in the challenging COCO object detection benchmark. A common pitfall is that some people believe Deformable DETR is "fast". Deformable DETR is indeed fast in terms of the convergence, *i.e.*, the total training epochs required by Deformable DETR are much smaller compared with the original DETR. However, Deformable DETR is not fast in terms of inference FPS, which is even slower than DETR. ViDT addresses the latency issues while maintaining a very high detection AP. It shows that Detection Transformer family also has the potential to be as strong as the highly-optimized detectors, *e.g.*, the YOLO series.

- This paper proposes an efficient Detection Transformer design principle that can generalize well and inspire future Detection Transformer design, *i.e.*, the encoder part in current DETR & Deformable DETR design results in high latency. Inspired by YOLOS, this paper proposes to incorporate [DET] tokens in the earlier stage of the model, which can compensate for the effects of the encoder while maintaining high inference FPS. Therefore the Transformer encoder part can be ablated by adding [DET] tokens to stage-4 of the backbone. Meanwhile, this paper finds that the Transformer decoder part is crucial for fast convergence for several Detection Transformer instances, which cannot be easily removed.

- This paper conducts a lot of experiments studying different configurations of DETR & Deformable DETR with ViT backbone, YOLOS, and the proposed ViDT in the same testbed. These results are quite meaningful and can be served as valuable references & baselines to the community as well as future work.



#### Weaknesses

- I believe there is one missing detail about the model design. If the [DET] x [PATCH] attention starts at stage-4, I wonder what happens to the [DET] x [DET] attention? Is [DET] x [DET] attention performed across all previous stages? And what are the feature dimensions of [DET] tokens? Are the feature dims always the same as the stage-4 feature, or they are gradually raising during different stages. In a word, I think the details and configurations of [DET] x [DET] attention need to be clarified.


--- Post rebuttal update ---
After reading the rebuttal, other reviewers' comments, and the discussion, overall I think the detector presented in the paper is strong in terms of both precision and efficiency, and the advantages I mentioned above are valuable, especially the first one. I keep my original rating.

**Summary Of The Paper:**

This paper proposes ViDT, a high-performance Detection Transformer with an impressive accuracy-speed trade-off. A lot of experiments as well as ablation studies are conducted to prove the effectiveness of the proposed detector. The design principle of ViDT can also generalize and inspire future detector design. Moreover, this paper also includes an in-depth analysis of several current Detection Transformer architectures.


**Summary Of The Review:**

Overall, I believe this is a meaningful paper from the perspective of both object detection performance and research. It will also inspire the design of [DET] token based framework of other object- and region- level downstream tasks, *e.g.*, instance segmentation, video instance segmentation as well as panoptic segmentation.

---

> ### Author Response · Authors · 2021-11-20
> **Response to Reviewer JKxV**
>
> We appreciate the reviewer's constructive comments and positive feedback.
>
> ---
>
> >**C1.**  *I wonder what happens to the [DET]x[DET] attention. Is [DET]x[DET] attention performed across all previous stages? And what are the feature dimensions of [DET] tokens? Are the feature dims always the same as stage 4, or they are gradually raising during different stages? I think the details and configurations of [DET]x[DET] attention need to be clarified.*
>
> It is correct that  [DET]x[DET] attention is performed across all previous stages, and the feature dimension of [DET] tokens increases gradually like [PATCH] tokens. As for the patch token, its feature dimension is increased by concatenating nearby [PATCH] tokens in a grid. However, this mechanism is not applicable for [DET] tokens since we maintain the same number of [DET] tokens for detecting a fixed number of objects in a scene. Hence, we simply repeat a [DET] token multiple times along the feature dimension to increase its size. This allows [DET] tokens to reuse all the projection and normalization layers in Swin Transformer without any modification. We discuss the detailed configuration of [DET] tokens in Appendix A.2. As mentioned in the abstract, we will release our code to the public for reproducibility.
>
> ---

---

### Official Review · Reviewer_vxqH · 2021-11-02

**Correctness:** 3
**Technical Novelty And Significance:** 2
**Empirical Novelty And Significance:** 2
**Recommendation:** 6
**Confidence:** 4

**Main Review:**

Strengths
- Compared with other transformer-based detectors, the proposed detector is effective and efficient, which has a high performance with a fast speed.
- The comparison with other transformer-based detectors is written well and readable.

Weaknesses
- Though the proposed detector is effective and efficient, the novelty is limited. To my knowledge, Reconfigured Attention Module (RAM) and token-based knowledge distillation is not novel. Also, other techniques, such neck structure and auxiliary decoding loss, are not original, which are borrowed from DETR and Deformable DETR.
- There are some concerns about  Reconfigured Attention Module (RAM). First, there is little about the cross-attention between [DET] $\times$ [PATCH]. Could you provide more details about how to process [DET] $\times$ [DET] and [DET] $\times$ [PATCH] at once? Second, the proposed RAM is not novel and may not have great contribution to the performance. RAM only has cross-attention between [DET] $\times$ [PATCH] in the last stage. The function of this module is the same as the transformer decoder. For the rest stages,  there are two operations: [PATCH] $\times$ [PATCH] and [DET] $\times$ [DET]. [PATCH] $\times$ [PATCH] is a default operation in Swin Transformer. To my knowledge, [DET] $\times$ [DET] is not meaningful and may have no effect on performance. Could you provide an experiment that removing [DET] $\times$ [DET] from the RAM module except the last stage?
- In Table 4 (w. Neck), the detection performance drop largely if all the stages are not involved. In this setting (w. Neck), the transformer decoder also have interaction between [DET] $\times$ [PATCH]. Thus, the performance drop is not only caused by the lack of interaction between [DET] $\times$ [PATCH] in RAM. Could you provide an experiment that only interacting with {4} [PATCH] in the transformer decoder?
- In Table 2, there is not an experiment about Deformable DETR + Swin-base. Could you provide this experiment?

**Summary Of The Paper:**

In this paper, the authors propose an efficient transformer-based detector. Specifically,  the proposed detector introduce Reconfigured Attention Module (RAM) , encoder-free neck and token-based knowledge distillation to boost performance with low computational overhead.

**Summary Of The Review:**

I appreciate the effective and efficient of the proposed detector, but I prefer to reject this paper for the above weaknesses. If the authors solve my concerns, I would like to raise my rate.

---

> ### Author Response · Authors · 2021-11-20
> **Response to Reviewer vxqH (1/3)**
>
> We appreciate the valuable and reasonable concerns by Reviewer vxqH. We hope that they can be resolved through our clarifications in this response and the revised paper.
>
> ---
>
> **Response to all the raised concerns**
>
> ---
>
> >**C1.** *The novelty is limited. To my knowledge, Reconfigured Attention Module (RAM) and token-based knowledge distillation is not novel, and other techniques, such as neck structure and auxiliary decoding loss, are borrowed from Deformable DETR.*
>
> ViDT addresses two major challenges in ViT and DETR for efficient and effective object detection.
>
>
> **(1) Efficient Attention Module.** YOLOS highly relies on the canonical ViT (e.g., DeiT), which suffers from its quadratic computational complexity to the number of [PATCH] tokens. Although Swin Transformer is much more efficient than DeiT (linear complexity to the number of tokens), adopting its efficient principle to YOLOS is challenging. Note that YOLOS receives the [DET] + [PATCH] tokens and *the number of  [DET] tokens of YOLOS must be the same in all stages*. Swin Transformer, on the other hand, gradually reduces the number of tokens across the stages, which makes Swin Transformer not applicable to YOLOS. Thus, we propose a Reconfigured Attention Module (RAM) that decomposes 'global self-attention to the [PATCH] tokens with [DET] appended' to 'the *three* different attention operations,' namely [PATCH]x[PATCH], [DET]x[DET], and [DET]x[PATCH] attention. With this decomposition, RAM makes direct use of the Swin Transformer's efficient mechanism only to [PATCH]x[PATCH] attention, the computationally heaviest part due to # [DET] $<<$ # [PATCH]. Thanks to the proposed RAM, ViDT maintains a fixed scale of [DET] tokens for object detection but constructs hierarchical representations starting with small-sized image patches for [PATCH] tokens.
>
> **(2) Insight on Neck Components.** (Deformable) DETR achieves competitive AP on the COCO benchmark, but its neck decoder results in a high latency in detection. Inspired by YOLOS, we incorporate [DET] tokens in the ViT backbone to make it to be a *standalone* object detector, which can compensate for the effect of the neck encoder. Therefore, the computationally heaviest part (i.e., neck encoder) is successfully ablated from the detection pipeline without compromising detection accuracies. In addition, we find that the neck decoder cannot be easily removed unlike the neck encoder because it is crucial for a fast convergence and a high AP.
>
> While each component (e.g. RAM, Neck-free Decoder) may be seemingly simple, the design principle of ViDT is novel, and indeed realizes a highly efficient and effective fully Transformer-based object detector. As commented by Reviewer JKxV, this work inspires further Transformer-based network designs.
>
> We are not sure the papers that make the reviewer think RAM and token-based knowledge distillation are not novel.  We would like to address concerns about the issue when we know the papers that the reviewer refers to.
>
>
> ---
>
>
> >**C2.** *Could you provide more details about how to process [DET]x[DET] and [DET]x[PATCH] attention at once?*
>
> Binding the two attention mechanisms is simple in terms of the implementation complexity. [DET]x[DET] and [DET]x[PATCH] attention generates a new [DET] token, which aggregates relevant contents in [DET] and [PATCH] tokens, respectively. Since the two attentions share exactly the same [DET] query embedding (obtained after the projection as shown in Figure 3), they can be processed at once by performing the scaled-dot product between $\text{[DET]}_Q$ and $\big[\text{[DET]}_K, \text{[PATCH]}_K\big]$ embeddings, where $Q$, $K$ are the key and query, and $[\cdot]$ is the concatenation.
> Then, the obtained attention map is applied to the $\big[\text{[DET]}_V, \text{[PATCH]}_V\big]$ embeddings, where $V$ is the value and $d$ is the embedding dimension.
>
> $i.e., \text{[DET]}_{new} = \text{Softmax}\big(\frac{\text{[DET]}_Q \big[\text{[DET]}_K, ~\text{[PATCH]}_K\big]^{\top}}{\sqrt{d}})\big[\text{[DET]}_V, \text{[PATCH]}_V\big]$.
>
> This is a widely used implementation in the recent Transformer-based architectures, such as YOLOS. We have included the discussion for [DET] tokens in Appendix A.2 in the revised paper. We will also make our code public upon acceptance.
>
>
> ---

---

> > ### Author Response · Authors · 2021-11-20
> > **Response to Reviewer vxqH (2/3)**
> >
> > ---
> >
> > >**C3.** *The proposed RAM is not novel and may not have a great contribution to the performance, since RAM only has cross-attention between [DET] x [PATCH] attention in the last stage? The function of this module is the same as the transformer decoder.*
> >
> > RAM has similar attention operations as the Transformer decoder at the neck. However, the main difference between RAM and the transformer decoder is that RAM makes the existing ViT backbone to be a standalone object detector, which means the [DET]x[PATCH] attention affects [PATCH]x[PATCH] attention to extract a more suitable representation for detection in a synergistic manner. Therefore, we note that the improvements with RAM are remarkable although ViDT maintains [DET]x[PATCH] cross-attention at the last stage.
> >
> > As shown in the table below, Deformable DETR is good w.r.t AP but shows very high latency due to the neck encoder component. If we simply remove the neck encoder, its AP drops drastically (-9.1) since fine-grained representation for object detection is difficult to obtain directly from the raw ViT backbone without using an additional neck encoder. However, ViDT compensates for the effect of the neck encoder by adding [DET] tokens into the body (backbone), thus successfully removing the computational bottleneck without compromising AP; it maintains +6.4 higher AP compared with the neck encoder-free Deformable DETR (the second row) while achieving similar FPS. Therefore, we conjecture that RAM contributes significantly to the performance w.r.t AP and FPS, especially for the trade-off between them. We included this discussion in Appendix C.2 and Table 14 of the revised paper.
> >
> >
> > | Method | Neck Encoder | Backbone | Epochs | AP |AP_S | AP_M | AP_L | Params | FPS |
> > | :-----: | :-----: | :-----: | :-----: | :-----: | :-----: | :-----: | :-----: | :-----: | :-----: |
> > | `Deformable DETR` | O | Swin-nano | 50 | 43.1 | 25.9 | 45.2 | 59.4 | 17M | 7.0 |
> > | `Deformable DETR` | X | Swin-nano | 50 | 34.0  | 18.0 | 36.3 | 18.4 | 14M | 22.4 |
> > | `ViDT` |X | Swin-nano | 50  | **40.4 (+6.4)**  | 23.2 | 42.8  | 55.9 | 16M | **20.0 (-2.4)** |
> >
> > ---
> >
> > >**C4.** *To my knowledge, [DET]x[DET] is not meaningful and may have no effect on performance. Could you provide an experiment that removes [DET]x[DET] from the RAM module except the last stage?*
> >
> > As suggested, we conduct an ablation study by removing the [DET]x[DET] attention one by one from the bottom stage with results summarized in the table below. When all the [DET]x[DET] attention is removed (the last row), the AP drops by 0.7, which is a considerable performance degradation. On the other hand, as long as the self-attention is activated at the last two stages, all the strategies exhibit similar AP. Therefore, only keeping [DET]x[DET] self-attention at the last two stages can further increase FPS (+0.2) without any degradation in AP. This observation could be used as another design choice for the AP and FPS trade-off. Therefore, we believe that [DET]x[DET] attention is meaningful to use in RAM.
> >
> > | Stage 1 |  Stage 2 |  Stage 3 |  Stage 4 | Backbone | Epochs | AP | FPS  |
> > | :-----: | :-----: | :-----: | :-----: | :-----: |  :-----: |  :-----: |  :-----: |
> > | O|O| O| O|  Swin-nano | 50 | 40.4 | 20.0 |
> > | |O| O| O|  Swin-nano | 50 | 40.3 | 20.1 |
> > | | | O| O| Swin-nano | 50 | 40.4 | 20.2 |
> > | | | | O|  Swin-nano | 50 | 40.1 | 20.3 |
> > | | | | | Swin-nano | 50 | 39.7 | 20.4 |
> >
> > ---

---

> > > ### Author Response · Authors · 2021-11-20
> > > **Response to Reviewer vxqH (3/3)**
> > >
> > > ---
> > >
> > > >**C5.** *The transformer decoder also has interaction between [DET]x[PATCH] attention. Thus, the performance drop is not only caused by the lack of interaction between [DET]x[PATCH] in RAM. Could you provide an experiment that only interacts with [PATCH] in the body?*
> > >
> > > The performance drop is not only caused by the lack of interaction between [DET]x[PATCH]. It is also associated with the use of spatial positional encoding, which greatly benefits object detection. As shown in Section 4.2.1, the spatial positional encoding helps in significant performance improvement, e.g., 5.0AP in Table 3 in our paper, or 7.8AP in Table 8 of the DETR paper.
> > >
> > > In Deformable DETR, this positional encoding is only injected at the neck Transformer encoder, but ViDT removes this encoder due to its computational bottleneck for an efficient detector. To remedy this problem, we instead inject the positional encoding into the features associated with cross-attention in RAM, as can be seen from the left side of Figure 3. Therefore, if all the cross-attention is deactivated, there are two reasons for the performance degradation: (1) missing-cross attention (the lack of information change between [DET] and [PATCH] tokens) and (2) missing spatial encoding.
> > >
> > > In addition, we have conducted the experiment -- interacting [PATCH] with [DET] only in the Transformer decoder (i.e., Swin-Transformer w.o. RAM + Transformer Decoder). The results are shown on the second row of the table for C3 above. In this case, the AP is only 34.0 due to the absence of the neck encoder. However, ViDT complements the effect of the neck encoder by appending [DET] tokens into the Swin Transformer backbone, achieving 40.4AP. Furthermore, if we only remove the sinusoidal positional encoding for the cross-attention in RAM, it shows 39.2AP, which is 1.2AP lower than ViDT.
> > >
> > > ---
> > >
> > > >**C6.** There is not an experiment about Deformable DETR + Swin-base.
> > >
> > > We agree that (Deformable) DETRs with base models are good references to readers, although their performances are not good in terms of AP and FPS. During the rebuttal period, we trained all the compared methods with DeiT- and Swin-base models, as summarized in the table below.  We have added all the results in Table 2 of the revised paper. Note that DETR models show catastrophically slow FPS compared to our ViDT models with a similar parameter size.
> > >
> > > | Method  | Backbone | Epochs | AP |AP_S | AP_M | AP_L | Params | FPS (Single) | FPS (Batch) |
> > > | :-----: | :-----: | :-----: | :-----: | :-----: | :-----: | :-----: | :-----: | :-----: | :-----: |
> > > | `DETR` | DeiT-base | 50 | 37.1 | 14.7 | 39.4 | 52.9 | 0.1B | 4.3 | 4.9 |
> > > | `DETR` | Swin-base | 50 | 40.7 | 18.3 | 44.1 | 62.4 | 0.1B | 9.7 | 12.6 |
> > > | `Deformable DETR` | DeiT-base | 50 | 46.4 | 26.7 | 50.1 | 65.4 | 0.1B | 4.4 | 5.3 |
> > > | `Deformable DETR` | Swin-base | 50 | 51.4 | 34.5 | 55.1 | 67.5 | 0.1B | 4.8 | 5.4 |
> > >
> > > ---

---

> ### Author Response · Authors · 2021-11-26
> **Please let us know whether you have additional questions**
>
> Dear Reviewer vxqH,
>
> We appreciate your comments. We have provided more results/explanations based on your review. Please go over our response and let us know whether you have additional questions or not.
>
> Thank you,

---

> > ### Comment · Reviewer_vxqH · 2021-11-27
> > **Reviewer Response to Author Response**
> >
> > Thanks for the great efforts in the rebuttal. The authors have resolved most of my concerns, but I still have a concern about C5. I need to clarify my question: could you provide an experiment that only interacting with **stage 4** [PATCH] in the transformer decoder? In the transformer decoder, there is a cross-attention between [DET] and [PATCH] of all stages. For early stages, the features may have little useful information for detection. With this large [PATH] set, it is hard to extract useful information. This is caused by softmax function, which produces positive value for each [PATH]. Maybe this is another reason for performance drop.

---

> > > ### Author Response · Authors · 2021-11-27
> > > **We would like to fully address your concern!**
> > >
> > > Dear Reviewer vxqH,
> > >
> > > Thank you for the clarification of C5. We would like to fully address your concerns, so could you explain your question in more detail?
> > >
> > > As you mentioned, there is a cross-attention for each layer in the Transformer decoder (we used **6 layers** of deformable transformer decoder).
> > >
> > > In the deformable decoding layer, it receives **four levels** of hierarchical [PATCH] tokens generated from each stage of Swin Transformer (i.e., stages 1, 2, 3, 4 at the Swin backbone). Since the same hierarchical [PATCH] tokens are provided to all layers in the decoder, **the information in the [PATCH] tokens is exactly the same in all the decoding layers (note that there is no [PATCH]x[PATCH] attention in the decoder)**.  That is, the decoding layer aggregates multi-scale [PATCH] tokens to generate new [DET] tokens via cross-attention without affecting the [PATCH] information; it only decides which multi-scale [PATCH] tokens should be aggregated for better [DET] tokens.
> > >
> > > Your question "only interacting with stage 4 [PATCH] in the transformer decoder" means that only providing [PATCH] tokens at Stage 4 of the Swin backbone as the input to the transformer decoder? (i.e., single-scale cross attention with stage 4 [PATCH] tokens).
> > >
> > > Please confirm that our understanding is correct.
> > >
> > > If this is correct, we do not benefit from using multi-scale features for object detection, which is crucial for detecting small-size objects in the scene. We will provide more results if our understanding is correct.
> > >
> > > Thanks,

---

> > > > ### Comment · Reviewer_vxqH · 2021-11-27
> > > > **Reviewer Response to Author Response**
> > > >
> > > > Yes, your understanding is correct.

---

> > > > > ### Author Response · Authors · 2021-11-27
> > > > > **Thank you for checking!**
> > > > >
> > > > > Thank you for checking!
> > > > >
> > > > > In this case, we carefully expect that the detection performance for small objects will rather drops drastically since we only use h/32 x w/32 (row resolution) [PATCH] tokens in the decoder.
> > > > >
> > > > > We will provide the exact results for this until Nov 29th! (Training takes a few days)
> > > > >
> > > > > Thanks,

---

> > > ### Author Response · Authors · 2021-11-29
> > > **An experiment that only interacts with stage 4 [PATCH] in the transformer decoder**
> > >
> > > Dear Reviewer vxqH,
> > >
> > > Thank you for considering another aspect that could be contributing to performance degradation; too many [PATCH] tokens at the early layer in the decoder make it hard to extract useful information when conducting cross-attention.
> > >
> > > To fully address your concern, we have conducted an experiment that only interacts with stage 4 [PATCH] tokens in the transformer decoder (i.e., **multi-scale** cross-attention -> **single-scale** cross-attention). We have tested four combinations to investigate the performance change when combined with RAM (the cross-attention in the backbone), as follows:
> > >
> > > (1) RAM: Cross-attention at Stage 4  	/ Decoder: Cross-attention with multi-scale [PATCH] tokens
> > >
> > > (2) RAM: Cross-attention at Stage 4 	/ Decoder: Cross-attention with single-scale Stage 4 [PATCH] tokens
> > >
> > > (3) RAM: No Cross-attention 				/ Decoder: Cross-attention with multi-scale [PATCH] tokens
> > >
> > > (4) RAM: No Cross-attention 				/ Decoder: Cross-attention with single-scale Stage 4 [PATCH] tokens
> > >
> > >
> > > |#| RAM| Decoder | Epochs | AP |AP_S | AP_M | AP_L |
> > > | :-----: | :-----: | :-----: | :-----: | :-----: | :-----: | :-----: |:-----: |
> > > | (1) | `{4}`  | `Multi-scale` 	| 50 | 40.4 | 23.2 | 42.5 | 55.8 |
> > > | (2) | `{4}`  | `Single-scale` | 50 | 30.3 (-10.1) | 14.6 (-8.6) | 32.8 (-9.7) | 45.9 (-9.9) |
> > >
> > >
> > > | # | RAM| Decoder | Epochs | AP |AP_S | AP_M | AP_L |
> > > | :-----: | :-----: | :-----: | :-----: | :-----: | :-----: | :-----: |:-----: |
> > > | (3) | `{ }`  | `Multi-scale` 	| 50 | 37.1 | 20.2 | 39.4 | 51.5 |
> > > | (4) | `{ }`  | `Single-scale` | 50 | 23.3 (-13.8) | 8.7 (-11.5) | 22.7 (-16.7) | 38.8 (-12.7)|
> > >
> > > Overall, only interacting single-scale (Stage 4) [Patch] tokens in the decoder rather drop APs significantly, which means that multi-scale cross-attention in the decoder is important to achieve high AP.  As you mentioned, if multi-scale features are used, the number of [PATCH] tokens is too many, but we found that **only a small number of sampled [PATCH] tokens participate in the deformable cross-attention regardless of the total number of [PATCH] tokens**. The number of sampled [PATCH] tokens $K$ (in the below Equation 1.) is set to be 4 for the Deformable attention.  Thus, only 8 (multi-head) x 4 (level of scale)  x 4 (K) = 128 [PATCH] tokens are used with multi-scale cross-attention, and only 8 (multi-head) x 1 (level of scale)  x 4 (K) = 32 [PATCH] tokens are used with single-scale cross-attention. These numbers are very small and constant regardless of the total number of tokens.
> > >
> > > Therefore, **we believe that using all the [PATCH] tokens from the four stages does not make it difficult to extract useful information**, and the benefit of using multi-scale features is very significant.
> > >
> > > `Equation 1. Deformable Cross-attention (Eq. (1) in the paper)`
> > >
> > > ${\rm MSDeformAttn}([\mathtt{DET}], \{ {x^{l}} \}_{l=1}^{L})$
> > >
> > > $=\sum_{m=1}^{M}{W_m}\Big[ \sum_{l=1}^{L} \sum_{k=1}^{K} A_{mlk}$ $ \cdot {W_m^{\prime}} {x^{l}}\big(\phi_{l}({p}) + \Delta {p_{mlk}}\big) \Big]$.
> > >
> > > $L$ is the level of scales
> > >
> > > $K$ is the total number of sampled keys for content aggregation
> > >
> > > $M$ is the number of heads
> > >
> > > $\phi_{l}({p})$ is the reference point of the $[\mathtt{DET}]$ token re-scaled for the $l$-th level feature map
> > >
> > > $\Delta{p_{mlk}}$ is the sampling offset for deformable attention
> > >
> > > $A_{mlk}$ is the attention weights of the $K$ sampled contents
> > >
> > > ${W_m}$ and ${W_m^{\prime}}$ are the projection matrices for multi-head attention
> > >
> > > Thanks,

---

> > > > ### Comment · Reviewer_vxqH · 2021-11-29
> > > > **Reviewer Response to Author**
> > > >
> > > > Thanks for the great efforts in the rebuttal. My concerns have been resolved, thus I will raise my score.

---

### Official Review · Reviewer_cEv4 · 2021-11-03

**Correctness:** 4
**Technical Novelty And Significance:** 2
**Empirical Novelty And Significance:** 2
**Recommendation:** 5
**Confidence:** 4

**Main Review:**

The description of the cross-attention is a bit informal and hard to understand. There is a lot going on in Fig 3 which could be formally described for the readers benefit.

YOLOS is not scalable because of global attention between [PATCH] tokens and [DET] tokens, however here the same issue arises with cross-attention (p4 last para) and is resolved by only considering global attention at the last stage. Could something similar be done with YOLOS? e.g. removing the global cross attention between patch and det in earlier stages. And if such global attention is removed in earlier stages, Swin type layers can also be used if I am not mistaken? In that case does the network correspond to YOLOS largely (without the neck?). This should be discussed as well.

DETR (vanilla and deformable) results are not given with the base models. I understand that the FPS would be low, but the results should be reported. In addition the DeiT-tiny and DeiT-small should also be the distillation based training versions. Since they are +2 points better than the non distillation ones, one could suspect that the (a) Deformable DETR + DeiT-tiny will improve from 39.2 to ~41, which would be then higher than (b) ViDT+Swin-nano (which is comparable at 16M params vs. 18M for (a)). This would also be fair wrt to backbone, from Tab1 and Sec4(Algothims) discussion, it is mentioned that the DEIT tiny, small are eqv. to Swin nano and tiny.

I am also wondering why are DETR(vanilla or Deformable) results with any base model (DeiT-base or Swin-base) not given in Tab2. While they might not be comparable wrt FPS, they should be given.

Similarly, the results with Conv backbones should also be given, since there are higher reported results in the literature. I wouldn't penalize the paper for not having "state-of-the-art" results. This request is so that the reader can be aware of the holistic landscape.


--- Post rebuttal update ---
I am not entirely convinced that a simpler baseline improvement to YOLOS can not be devised. But the proposed method is surely one way to go forward. The results and discussions could also be valuable to the readers. I am not upgrading my rating but I wouldn't argue strongly for rejections, specially when there is some support from the other reviewers.

**Summary Of The Paper:**

The paper builds upon the recent advances in transformer based image classification methods (ViT variants) and detection methods (DETR variants). It argues that naively replacing the conv feature backbone in DETR with a ViT based one, is problematic due to (i) the quadratic complexity of the self attention module in ViT, and (ii) then again the attention module in the transformer encoder decoder part (which they call "neck"). To remedy, it proposes to build on top of Swin Transformer backbone, by proposing a novel reconfigured attention module (RAM), and further removing the encoder-decoder in the neck, replacing it with only a lightweight decoder.

**Summary Of The Review:**

The paper is generally well written, however one of the main contribution, the reconfigurable attention module, should be more precisely and formally described, at least an appendix. Otherwise it is a little hard to get with Fig3 largely.

The contextualization of the paper wrt YOLOS is also something that could be improved. It seems that the model is more closely related to YOLOS than seems at first read.

The experiments should be made more complete with fair comparison for baseline backbones etc. And the results with conv backbones should also be given for completeness and benefit of the reader.

---

> ### Author Response · Authors · 2021-11-20
> **Response to Reviewer cEv4 (1/3)**
>
> We appreciate your valuable comments and suggestions. We hope that the concerns can be resolved through our clarifications in this response and the revised paper.
>
> ---
>
> >**Main novelty of the method**
>
> ViDT addresses two major challenges in ViT and DETR for efficient and effective object detection.
>
> **(1) Efficient Attention Module.** YOLOS highly relies on the canonical ViT (e.g., DeiT), which suffers from its quadratic computational complexity to the number of [PATCH] tokens. Although Swin Transformer is much more efficient than DeiT (linear complexity to the number of tokens), adopting its efficient principle to YOLOS is challenging. Note that YOLOS receives the [DET] + [PATCH] tokens and *the number of  [DET] tokens of YOLOS must be the same in all stages*. Swin Transformer, on the other hand, gradually reduces the number of tokens across the stages, which makes Swin Transformer not applicable to YOLOS. Thus, we propose a Reconfigured Attention Module (RAM) that decomposes 'global self-attention to the [PATCH] tokens with [DET] appended' to 'the *three* different attention operations,' namely [PATCH]x[PATCH], [DET]x[DET], and [DET]x[PATCH] attention. With this decomposition, RAM makes direct use of the Swin Transformer's efficient mechanism only to [PATCH]x[PATCH] attention, the computationally heaviest part due to # [DET] $<<$ # [PATCH]. Thanks to the proposed RAM, ViDT maintains a fixed scale of [DET] tokens for object detection but constructs hierarchical representations starting with small-sized image patches for [PATCH] tokens.
>
> **(2) Insight on Neck Components.** (Deformable) DETR achieves competitive AP on the COCO benchmark, but its neck decoder results in a high latency in detection. Inspired by YOLOS, we incorporate [DET] tokens in the ViT backbone to make it to be a *standalone* object detector, which can compensate for the effect of the neck encoder. Therefore, the computationally heaviest part (i.e., neck encoder) is successfully ablated from the detection pipeline without compromising detection accuracies. In addition, we find that the neck decoder cannot be easily removed unlike the neck encoder because it is crucial for a fast convergence and a high AP.
>
> While each component (e.g. RAM, Neck-free Decoder) may be seemingly simple, the design principle of ViDT is novel, and indeed realizes a highly efficient and effective fully Transformer-based object detector. As commented by Reviewer JKxV, this work inspires further Transformer-based network designs.
>
> ---

---

> > ### Author Response · Authors · 2021-11-20
> > **Response to Reviewer cEv4 (2/3)**
> >
> > **Response to all the raised concerns**
> >
> > ---
> >
> > > **C1.** *YOLOS is not scalable because of global attention between [PATCH] and [DET] tokens, however, here the same issue arises with cross-attention.*
> >
> > Yes, YOLOS is not scalable because it performs a single global attention for the [PATCH] tokens with [DET] tokens appended. The computational complexity of this attention is $\mathcal{O}(d^2({\sf P} + {\sf D}) + d({\sf P} + {\sf D})^2)$, which is *quadratic* to the number of [PATCH] tokens since # [DET] $<<$ # [PATCH], where ${\sf {P}}$ and ${\sf D}$ are the number of [PATCH] and [DET] tokens, respectively, and $d$ is the embedding dimension. For example, at the first stage, ${\sf P}$ and ${\sf D}$ are 66,650 and 100, respectively.
> >
> > However, cross-attention in our ViDT does not suffer from the scalability issue as *ViDT has a linear complexity to the number of the input patch tokens instead of quadratic complexity*. The proposed RAM decomposes the single global attention in YOLOS into the three different attention operations, (1) $[\mathtt{PATCH}]\times[\mathtt{PATCH}]$ local self-attention with window partition, $\mathcal{O}(d^2 {\sf P} + d k^2 {\sf P})$; (2) $[\mathtt{DET}]\times[\mathtt{DET}]$ global self-attention, $\mathcal{O}(d^2 {\sf D} + d {\sf D}^{2})$; (3) $[\mathtt{DET}]\times[\mathtt{PATCH}]$ global cross-attention, $\mathcal{O}(d^2 ({\sf D} + {\sf P}) + d {\sf D} {\sf P})$, where $k$ is the window size ($k << {\sf P}, {\sf D}$).  In total, the computational complexity of RAM is $\mathcal{O}(d^2 ({\sf D} + {\sf P}) + dk^2 {\sf P} + d {\sf D}^{2} + d {\sf D} {\sf P})$, which is *linear* to the number of [PATCH] tokens because the local attention in Swin Transformer is applied for [PATCH]x[PATCH] attention. Please see the detailed complexity analysis in Appendix A.1, which is newly added in the revised paper.
> >
> > The main complexity difference between YOLOS and ViDT comes from the [PATCH]x[PATCH] attention, i.e., *quadratic*: $\mathcal{O}(d^2 {\sf P} + d {\sf P}^2)$ in YOLOS v.s. *linear*: $\mathcal{O}(d^2 {\sf P} + d k^2 {\sf P})$ in ViDT.  In particular, the gap is significantly large at the first stage where ${\sf P}$ is 66,650. In contrast, the complexity of [DET]x[PATCH]  attention, $\mathcal{O}(dDP)$, is not the major part because it is linear to the number of [PATCH] tokens. Therefore, the quadratic complexity issue does not occur in RAM.
> >
> > ---
> >
> > >**C2-1.** *Could something similar be done with YOLOS?, e.g., removing the cross-attention between [PATCH] and [DET].*
> >
> > As discussed in C1, the computational bottleneck of YOLOS is the global [PATCH]x[PATCH] self-attention with quadratic complexity. Thus, removing the cross-attention between [DET] and [PATCH] tokens cannot make YOLOS have a linear complexity similar to ViDT, and also this change will negatively affect the detection accuracy of YOLOS.
> >
> > ---
> >
> > >**C2-2.** *If such global cross-attention between [DET] and [PATCH] is removed in an earlier stage in YOLOS, Swin type layers can also be used if I am not mistaken? Then, does the proposed ViDT correspond to YOLOS largely (without the neck)?*
> >
> > YOLOS receives the append [DET] and [PATCH] tokens as the input. Here, the number of  [DET] tokens must be same across all the stages, so the Swin Layer cannot be applied to YOLOS because it gradually changes the number of tokens. That makes it challenging to incorporate  Swin's mechanism into YOLOS. The key idea in RAM for this challenge is decomposing the original attention into the proposed three different attentions and applying the Swin mechanism only into the [PATCH]x[PATCH] attention. Therefore, the proposed ViDT does not correspond to YOLOS.
> >
> > The purpose of performing the cross-attention only at the last stage in ViDT is to make it more efficient. Although it has a linear complexity to the patch size, it could make the model slow when the number is extremely high, i.e., 66,650 tokens at the first stage.
> >
> > ---

---

> > > ### Author Response · Authors · 2021-11-20
> > > **Response to Reviewer cEv4 (3/3)**
> > >
> > > ---
> > >
> > > >**C3.** *DeiT-tiny and DeiT-small should also be the distillation-based training versions. This would be fair wrt to backbone because they are +2 points (ImageNet accuracy) better than the non-distillation ones.*
> > >
> > > We agree that using the distilled version of DeiT-tiny and -small models is a fair comparison. We attach the detailed comparisons below. When replacing the scratch-trained backbones to the distillation-based backbones, there is an improvement in AP of $0.9-2.7$ but their FPS is significantly worse than ViDT due to (1) the quadratic complexity of DeiT and (2) the neck encoder in (Deformable) DETR. Figure 1, Table 1, and Table 2 in the paper have been updated with the results of the distilled DeiT models.
> > >
> > > As pointed out by the reviewer, Deformable DETR (DeiT-tiny) improves from 39.2 to 40.8 (0.4 higher than ViDT (Swin-nano)). However, Deformable DETR (DeiT-tiny dist.) shows drastically slow (almost one third in the batch inference scenario) FPS compared to ViDT (Swin-nano); 16.3 FPS (Batch) for DeiT-tiny dist. vs. 45.8 FPS (Batch) for ViDT; and 12.4 FPS (Single) vs. 20.0 FPS (Single), where FPS (Single) and FPS (Batch) refer to inference with batch size 1 and 4, respectively.
> > >
> > >
> > > | Method | Backbone | Epochs | AP |AP_S | AP_M | AP_L | Params | FPS (Single) | FPS (Batch) |
> > > | :-----: | :-----: | :-----: | :-----: | :-----: | :-----: | :-----: | :-----: | :-----: |  :-----: |
> > > | `DETR` | DeiT-tiny | 50 | 29.1 |9.2 | 29.3 | 49.5 | 24M | 10.9 | 13.1 |
> > > | `DETR` | DeiT-tiny (dist.)| 50 | 30.0 | 9.9 | 30.8 | 50.6 | 24M | 10.9 | 13.1 |
> > > | `DETR` | DeiT-small | 50 | 30.8 | 10.5 | 31.0 |52.1 | 39M |  7.8 | 8.8 |
> > > | `DETR` | DeiT-small (dist.)| 50 | 32.4 | 11.3 | 33.5 |53.7 | 39M |  7.8 | 8.8 |
> > > | `Deformable DETR` | DeiT-tiny | 50 | 39.2 | 19.5 |41.5 | 58.0 | 18M | 12.4 | 16.3 |
> > > | `Deformable DETR` | DeiT-tiny (dist.)| 50 | 40.8 | 21.4 |43.4 | 58.2 | 18M | 12.4 | 16.3 |
> > > | `Deformable DETR` | DeiT-samll | 50 | 40.9 | 20.4 | 43.8 | 59.6 | 35M | 8.5 | 10.2 |
> > > | `Deformable DETR` | DeiT-samll (dist.)| 50 | 43.6 | 23.3 | 47.1 | 62.1 | 35M | 8.5 | 10.2 |
> > > | `ViDT` | Swin-nano | 50 | 40.4 | 23.2 | 42.5 | 55.8 | 16M | 20.0 | 45.8 |
> > > | `ViDT` | Swin-tiny | 50 | 44.8 | 25.9 | 47.6 | 62.1 | 38M | 17.2 | 26.5 |
> > >
> > > ---
> > >
> > > >**C4.** *DETR (vanilla and deformable) results are not given with the base models (DeiT-base and Swin-base).*
> > >
> > > We agree that (Deformable) DETRs with base models are good references although they do not perform well in terms of AP and FPS. During the rebuttal period, we train all the compared methods with DeiT- and Swin-base models, as summarized in the table below.  We have added all the results in the Table 2 of the revised paper. Note that DETR models have very slow FPS compared to our ViDT models with a similar model size.
> > >
> > > | Method  | Backbone | Epochs | AP |AP_S | AP_M | AP_L | Params | FPS (Single) | FPS (Batch) |
> > > | :-----: | :-----: | :-----: | :-----: | :-----: | :-----: | :-----: | :-----: | :-----: | :-----: |
> > > | `DETR` | DeiT-base | 50 | 37.1 | 14.7 | 39.4 | 52.9 | 0.1B | 4.3 | 4.9 |
> > > | `DETR` | Swin-base | 50 | 40.7 | 18.3 | 44.1 | 62.4 | 0.1B | 9.7 | 12.6 |
> > > | `Deformable DETR` | DeiT-base | 50 | 46.4 | 26.7 | 50.1 | 65.4 | 0.1B | 4.4 | 5.3 |
> > > | `Deformable DETR` | Swin-base | 50 | 51.4 | 34.5 | 55.1 | 67.5 | 0.1B | 4.8 | 5.4 |
> > >
> > > ---
> > >
> > > >**C5.** *The results with Conv backbones should also be given so that readers can be aware of the holistic landscape.*
> > >
> > > To show the holistic landscape, we have included the results with Conv backbones (ResNet-50), which are summarized in the table below. The AP values are borrowed from (Deformable) DETR papers, but FPS is calculated in our environment with a single NVIDIA V100 GPU. ViDT (Swin-tiny) achieves +3.5 (Single) and +7.1 (Batch) higher FPS than Deformable DETR (ResNet-50). In addition, it achieves AP higher than vanilla DETR with much smaller training epochs. Please see Appendix C.1 for more details.
> > >
> > > | Method  | Backbone | Epochs | AP |AP_S | AP_M | AP_L | Params | FPS (Single) | FPS (Batch) |
> > > | :-----: | :-----: | :-----: | :-----: | :-----: | :-----: | :-----: | :-----: | :-----: | :-----: |
> > > | `DETR` | ResNet-50 | 500 | 42.0 | 20.5 | 45.8 | 61.1 | 41M | 22.8 | 38.6 |
> > > | `DETR-DC5` | ResNet-50 | 500 | 43.3 | 22.5 | 47.3 | 61.1 | 41M | 12.8 | 14.2 |
> > > | `DETR-DC5` | ResNet-50 | 50 | 35.3 | 15.2 | 37.5 | 53.6 | 41M | 12.8 | 14.2 |
> > > | `Deformable DETR` | ResNet-50 | 50 | 45.4 | 26.8 | 48.3 | 64.7 | 40M | 13.7 | 19.4 |
> > > | `ViDT` | Swin-tiny | 50  | 44.8 | 25.9 | 47.6 | 62.1 | 38M | 17.2 | 26.5 |
> > >
> > > ---
> > >
> > > >**C6.** *The reconfigurable attention module should be more precisely and formally described at least an appendix.*
> > >
> > > Thanks for the suggestion. We add more details of the reconfigured attention model in Section 3.1. In addition,  We discuss the computational complexity and further algorithmic design of RAM in Appendix A.
> > >
> > > ---

---

> > > > ### Comment · Reviewer_cEv4 · 2021-11-29
> > > > **Thank you for the clarifications and further results**
> > > >
> > > > Dear Authors,
> > > >
> > > > Sorry for the delay, and thank you for the clarifications.
> > > >
> > > > Although I still believe that alternate and simple modification to YOLOS algorithm could be done to improve the speed similar to the proposed method (which is perhaps the main improvement here), the method is one way of going forward. I might have been misunderstood a bit (regarding the global attention part), but the clarifications nonetheless do make things more clear.
> > > >
> > > > Thank you also for the complete results, it makes the full landscape clear, giving some advantage to the proposed method in terms of speed for comparable performances.
> > > >
> > > > Best,
> > > > Gaurav

---

> > > > > ### Author Response · Authors · 2021-11-29
> > > > > **Thank you for the feedback!**
> > > > >
> > > > > Dear Reviewer cEv4,
> > > > >
> > > > > We really appreciate your feedback on our response. We agree that YOLOS could be improved in another way to resolve its heavy computation. Hence, we believe that the proposed reconfigured attention module (RAM) will inspire those future works and can be used as a good reference and baseline. Please note that this is the first work to incorporate the efficient mechanism of Swin Transformer to improve the YOLOS.
> > > > >
> > > > > In addition, we would like to emphasize that **solving the slow performance of YOLOS is not everything for high detection performance**. As can be seen in the table below, the two neck-free detectors, YOLOS and its improved ViDT (w.o. Neck), shows relatively poor AP performance although ViDT (w.o. Neck) resolves the computational bottleneck in YOLOS.
> > > > >
> > > > > To achieve high AP and fast training speed, **the integration with detection transformers (i.e., the neck components) is necessary**. Therefore, we proposed to integrate only the neck decoder (DETR) with our proposed detection backbone, which is justified by the two new observations: (1) the computationally heaviest part (i.e., neck encoder) is successfully ablated from the detection pipeline without compromising detection accuracies and (2) the neck decoder cannot be easily removed, unlike the neck encoder. Therefore, this integrated version called ViDT improves AP significantly while achieving high FPS.
> > > > >
> > > > > **The main novelty of this work is to synergize vision and detection transformers for the best trade-off between accuracy and efficiency.** The proposed ViDT shows that Detection Transformer family also has the potential to be as strong as the highly-optimized detector.
> > > > >
> > > > > |Method | Backbone | Epochs | AP |AP_S | AP_M | AP_L | Params | FPS (Single) | FPS (Batch) |
> > > > > | :-----: | :-----: | :-----: | :-----: | :-----: | :-----: | :-----: |:-----: |:-----: |:-----: |
> > > > > | `YOLOS` | DeiT-tiny	| 150 | 30.4 | 12.4 | 31.8 | 48.2 | 6M | 28.1 | 31.3 |
> > > > > | `YOLOS` | DeiT-small| 150 | 36.1 | 15.6 | 38.4 | 55.3 | 30M | 9.3 | 11.8 |
> > > > > | `ViDT (w.o. Neck)` | Swin-nano  | 150 | 28.7 | 12.3 | 30.7 | 44.1 | 7M | 36.5 | 64.4 |
> > > > > | `ViDT (w.o. Neck)` | Swin-tiny  | 150 | 36.3 | 16.4 | 39.0 | 54.3 | 29M | 28.6 | 32.1 |
> > > > > | `ViDT`  | Swin-nano  | 50 | 40.4 | 23.2 | 42.5 | 55.8 | 16M | 20.0 | 45.8 |
> > > > > | `ViDT` | Swin-tiny  | 50 | 44.8 | 25.9 | 47.6 | 62.1 | 38M | 17.2 | 26.5 |
> > > > >
> > > > > Thank you again for the constructive feedback, and let us know whether you have additional questions or not.
> > > > >
> > > > > Best,

---

> ### Author Response · Authors · 2021-11-26
> **Please let us know whether you have additional questions**
>
> Dear Reviewer cEv4,
>
> We appreciate your comments. We have provided more results/explanations based on your review. Please go over our response and let us know whether you have additional questions or not.
>
> Thank you,

---

> > ### Author Response · Authors · 2021-11-28
> > **Dear Reviewer cEv4**
> >
> > We thank you for the precious review time and valuable comments/suggestions. We have provided corresponding responses and results, which we believe have covered your concerns: (1) detailed description for RAM including complexity analysis, (2) more fair DeiT models, (3) results with base models, and (4) comparison with CNN backbones.
> >
> > Please take a close look at our paper and response, and let us know whether you have additional questions or not.
> >
> > Thank you,

---

> > ### Author Response · Authors · 2021-11-29
> > **A gentle reminder for Reviewer cEv4**
> >
> > Dear Reviewer cEv4,
> >
> > We appreciate your constructive comments for helping us to improve our paper in many aspects. This is a gentle reminder since the final discussion stage ends soon.
> >
> > We would like to ask if Reviewer cEv4 may have any further questions regarding our submission so that we can still respond.
> >
> > Thanks,

---

### Author Response · Authors · 2021-11-20
**General Response**

Dear reviewers and meta reviewers,

We appreciate all the comments including 1) the proposed design principles can generalize well and inspire future work, and the results will be a good reference and baseline to the community (**Reviewer JKxV**); 2) the paper is generally well written and the reconfigure attention module is novel (**Reviewer cEv4**); 3) the proposed detector is effective and efficient and has a high performance with a fast speed (**Reviewer vxqH**).

In the following, we address the concerns from the reviewers. In addition, we include all the responses in the revised manuscript. For ease of reviewing, we highlight the added or revised text in **red color**.

We summarize the main changes as follows:

- Add a computational complexity analysis in Appendix A.1  (**C1, C2**  by  **Reviewer cEv4**)
- Replace the results of DeiT-tiny and DeiT-small backbones with those of their distillation versions in Figure 1, Table 1, and Table 2  (**C3**  by  **Reviewer cEv4**)
- Add the results with DeiT-base and Swin-base backbones for (Deformable) DETR in Figure 1 and Table 2  (**C4**  by  **Reviewer cEv4** and **C6**  by **Reviewer vxqH**)
- Add the comparison with CNN backbones in Appendix C.1 and Table 13 (**C5**  by  **Reviewer cEv4**)
- Add more detailed explanation for RAM in Section 3.1 and Appendix A  (**C6**  by  **Reviewer cEv4**)
- Add the detailed procedure for binding [DET]x[DET] and [DET]x[PATCH] attention in Appendix A.2.1 (**C2**  by  **Reviewer vxqH**)
- Add an ablation study for the use of [DET]x[DET] attention in Appendix C.4 and Table 15 (**C3**  by  **Reviewer vxqH**)
- Add the comparison with variations of existing pipelines in Appendix C.2 and Table 14  (**C4**  by  **Reviewer vxqH**)
- Add the detail of embedding dimensions for [DET] tokens in Appendix A.2.2 (**C1**  by  **Reviewer JKxV**)

Should you have any further questions or suggestions, please put your comments on OpenReview. We will address all the raised concerns according to the reviewing policy.

---

### Decision · Program_Chairs · 2022-01-20

**Decision:**

Accept (Poster)

**Comment:**

The paper introduces an object detection method that integrates vision and detection transformers through a novel Reconfigured Attention Module (RAM). Among other questions, the reviewers raised concerns about fair comparison with baselines, limited novelty of the RAM module, completeness of experiments, and missing details. The rebuttal adequately addressed these concerns with clarifications and additional experiments. R1 remained unconvinced that a simple modification to YOLOS could not be devised to improve the speed similar to the proposed method, but stated he/she wouldn’t argue strongly for rejection. While this is a legitimate concern, the AC agrees with R2 and R3 that the paper has enough merits to be accepted at ICLR, as the results are strong and are likely to have significant practical value.